# A new merge of global surface temperature datasets since the start of the 20th Century

Xiang Yun[1,2], Boyin Huang [3], Jiayi Cheng[1#], Weihui Xu[4], Shaobo Qiao[1#], Qingxiang Li [1,#*]

1 School of Atmospheric Sciences and Guangdong Province Key Laboratory for Climate Change and Natural Disasters, SUN Yat-Sen University, Guangzhou, China

2 Chinese Academy of Meteorological Sciences, CMA, Beijing, China

3 National Centers of Environmental Information, NOAA, Asheville, USA

4 National Meteorological Information Center, CMA, Beijing, China

# Southern Laboratory of Ocean Science and Engineering (Guangdong Zhuhai), Zhuhai, China

*Corresponding to: Qingxiang Li (liqingx5@mail.sysu.edu.cn)

## Abstract

Global surface temperature (ST) datasets are the foundation for global climate change research.

There are several global ST datasets developed by different groups in NOAA/NCEI,NASA/GISS and

UKMO Hadley Centre & UEA/CRU. This study presents a new global ST dataset, the China Merged

Surface Temperature (CMST) dataset. CMST is created by merging the China-Land Surface Air

Temperature (C-LSAT1.3) with the sea surface temperature (SST) data from the Extended

Reconstructed Sea Surface Temperature version 5 (ERSSTv5). The merge of C-LSAT and ERSSTv5

shows a high spatial coverage extended to the high latitudes and is more consistent with a reference of

multi-datasets average in Polar Regions. Comparisons indicate that CMST is consistent with other



existing global ST datasets in interannual-decadal variations and long-term trends at global, hemispheric,

and regional scales from 1900 to 2017. Therefore CMST dataset can be used for global climate change

assessment, monitoring, and detection. CMST dataset presented in this article is publicly available

at: https://doi.pangaea.de/10.1594/PANGAEA.901295 (Yun et al., 2019) and has been published on the

5  Climate Explorer website of the Royal Netherlands Meteorological Institute (KNMI) at:

http://climexp.knmi.nl/select.cgi?id=someone@somewhere&field=cmst.



# 1. Introduction

A common measure in observing the climate change is the long-term trend of the Global Mean Surface Temperatures (GMST). Therefore, the biases of the observed surface temperature (ST) dataset, especially the sampling bias of the stations at high latitudes, has received much attention (Cowtan and Way, 2014; Jones, 2016; Li et al., 2017; Simonds et al., 2017; Huang et al., 2017a). As a basis for climate change research and a verification benchmark for other climatic data products, the optimization and improvement of observational climate data are a long-term task.

The Intergovernmental Panel on Climate Change (IPCC, 2013) has exhibited four global land surface air temperature (LSAT) observation series and three global ST series. The four LSATs are the Climatic Research Unit (CRU) land surface air temperature, version 4 (CRUTEM4; Jones et al., 2012), Global Historical Climatology Network-monthly (GHCNm) temperature, version 3 (GHCNm v3; Lawrimore et al., 2011), Goddard Institute for Space Studies analysis of land surface air temperature (GISS; Hansen et al., 2010), and Berkeley Earth Surface Temperature group land temperature (Berkeley; Rohde et al., 2013). The three global ST series are the Met Office Hadley Centre and Climatic Research Unit Temperature version 4 (HadCRUT4; Morice et al., 2012), Merged Land–Ocean Surface Temperature (MLOST; Vose et al., 2012b), and Goddard Institute for Space Studies Surface Temperature Analysis (GISTEMP; Hansen et al., 2010).

These datasets all indicate that the Earth experienced a "warming hiatus" period over 1998-2012, which has attracted the attention of many researchers around the world. However, by analyzing the sea surface temperature (SST) and global ST in National Oceanic and Atmospheric Administration / National Centers for Environmental Information (NOAA/NCEI), Karl et al. (2015) suggested that the "warming hiatus" is due to the artifact of the data processing. Similarly, Lewandowsky et al. (2015) noted that this short-term warming trend "hiatus" is a conditional statistical artifact, not a real scientific fact. After correcting the sampling biases of the temperature data in the Arctic region, several studies reached the similar conclusion by using reanalysis data (Simonds et al., 2017), satellite remote sensing data (Cowtan and Way, 2014), and Arctic buoy data (Wang et al., 2017).

These global ST data products have been updated over the past several years since the publication of IPCC (2013). NOAA has updated the Extended Reconstructed Sea Surface Temperature (ERSST) version 3 to ERSSTv4 (Huang et al., 2015) and ERSSTv5 (Huang et al., 2017), updated LSAT dataset GHCNm v3 to GHCNm v4 (Menne et al., 2018), and renamed MLOST to NOAA Global Surface Temperature (NOAAGlobalTemp). GISTEMP has been updated its SST component to ERSSTv5 (Huang et al., 2017b). CRUTEM has been updated to CRUTEM4.6. The Met Office has updated the Hadley Centre SST to version 3 (HadSST3) using the median of 100 ensemble members. The Berkeley team uses the median of the ensembles of HadSST3 to form the Berkeley Earth Surface Temperature (BEST) dataset.



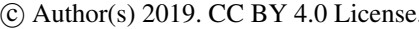

The updates of these products are based on the advanced knowledge of data analysis methodology or improved data availability. In general, the GMST has continuously been improved by the increased number and area coverage of observational data over the land (LSAT) and in the oceans (SST). There are two aspects to improve LSAT datasets: The first is to increase the density of stations and data coverage especially in the key areas with sparse observations. For example, the number of observations is increased in both C-LSAT (Xu et al., 2018) and GHCNm v4 (Menne et al., 2018) using newly released International Surface Temperature Initiative (ISTI) datasets (Thorne et al., 2011) or datasets through regional cooperation with Asian countries such as Vietnam and South Korea. The larger number of observations increases the coverage of datasets and therefore reduces the sampling biases, especially at high latitudes (Polar Regions) and in the observation-sparse regions such as South America and Africa. The second is to improve the accuracy of regional climate changes. For example, the latest C-LSAT (Xu et al., 2018) has integrated more regional homogenization results, especially in China (Li et al., 2009; Xu et al., 2013), East Asia, Europe, Australia (Trevin, 2013), and Canada (Vincent et al., 2012).

Similarly, there are two aspects to improve the SST datasets: The first is to integrate better raw observational data. For example, the ERSSTv5 uses the most recently available International Comprehensive Ocean-Atmosphere Data Set release 3.0 (ICOADS R3.0; Freeman et al., 2017), uses more accurate buoy data to adjust ship data, and uses optimized climate modes. The second is to replace

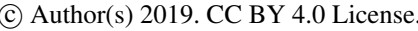

a single analysis with multi-member ensemble analyses. For example, HadSST3 introduces a variety of bias correction models and uses the median SST from the 100 ensemble members as the best estimate.

Among all the existing global ST datasets (for example, HadCRUT and NOAAGlobalTemp), the merging methods on combining the land and ocean datasets are basically similar. The merging process

of HadCRUT includes: First, the land and ocean data are processed into 100 ensemble datasets according to the bias evaluation parameters, and the anomaly values are calculated for each grid box separately. Then, anomalies in the grid boxes of the land and ocean boundary are weighted by the fraction of land and ocean areas. If the land area covers less than 25%, it is calculated as 25%. If there is a measured SAT anomaly in a grid box covered with sea-ice, The SAT will be used to represent the SST

anomaly. The HadCRUT ensemble datasets has a reference period of 1961-1990 and a resolution of $5°\times5°$ (Morice et al., 2012). The merging process in NOAAGlobalTemp includes three steps: First is the identification of the LSAT (or SST) low frequency changes by calculating the moving average of temperature anomaly data, and the identification of the LSAT (or SST) residual high frequency changes via the Empirical Orthogonal Teleconnection (EOT) modes. Then, the low- and high-frequency

components are integrated together. Finally, the SST data at $2°\times2°$ resolution is averaged into the grid at $5°\times5°$ resolution, and the land and ocean reconstructions are then merged into a global reconstruction similar to HadCRUT (Vose et al., 2012).

This study presents a new merged global ST dataset based on the recently developed C-LSAT and

the latest ERSSTv5 using a method similar to HadCRUT and NOAAGlobalTemp, which provides a

new reference to the climate/climate change studies. The remainder of this paper is arranged as follows:

The land and ocean datasets and their updates are briefly introduced in section 2; the merging process of

CMST is given in section 3; the comparisons of CMST with other existing ST datasets are discussed in

section 4; the availability of the resulting dataset (Yun et al., 2019) is reported in Sect. 5; and a summary

of results are presented in section 6.

## 2. Updates of land and ocean data

### 2.1 Land surface air temperature data

The C-LSAT1.0 dataset (Xu et al., 2018) processed SAT data since 1900 from a total of 14 data

sources, including three global data sources (CRUTEM 4.6, GHCNv3, and BEST), three regional data

sources from Scientific Committee on Antarctic Research (SCAR), Daily dataset for European Climate

Assessment (ECA&D), and Historical Instrumental climatological Surface Time series of the greater

Alpine region (HISTALP), and eight national data sources from China, America, Russia, Canada,

Australia, Korea, Japan, and Vietnam. Two steps have been taken to ensure the homogeneity of the

station time series: First, the data series from the existing national homogenized datasets have been

directly integrated into C-LSAT without any change, which are approximately 50% of the stations in

C-LSAT. Second, the inhomogeneities in the rest of the station series have been detected and adjusted

with the penalized maximal t-test method (Wang et al., 2007).

The C-LSAT version 1.3 is used in this study. Compared with C-LSAT version 1.0 from 1900 to 2014 (Xu et al., 2018), the data in version 1.3 have been updated to Dec 2017. According to Xu et al. (2018), national, regional and global datasets are ranked as higher, middle and lower priorities, respectively. Based on the priority of the data resources, 4917 stations with higher priority are added, and 1364 stations with lower priority are deleted. Most of the newly added raw data are from the International Surface Temperature Initiative (ISTI) Projects, and have been homogenized through the same approach as Xu et al. (2018). The distribution of these extra 3553 stations is shown in Figure 1. According to Xu et al. (2018), the C-LSAT version 1.0 had some advantages over the existing global LSAT datasets in station numbers and spatial coverage. Thus the current C-LSAT version 1.3 has more station numbers than the existing datasets in many regions over the global land surface. Figure 1 shows the extra stations compared with version 1.0 and Table 1 shows the comparison of the station numbers for different datasets, indicating enhanced coverage and distribution/sampling of LSAT observations.

For the comparison purposes, other LSAT datasets are collected from CRUTEM4 (https://www.metoffice.gov.uk/hadobs/), GHCNm v3 (ftp://ftp.ncdc.noaa.gov/pub/), Berkeley Earth (BE) (http://www.berkeleyearth.dev/). All of the above data sets were downloaded in August 2018. The following calculations are based on the stations with a time length greater than 15 years between 1900 and 2017.

From a global and hemispheric perspective, C-LSAT version 1.3 dataset has more stations than the other datasets in the Global and Southern Hemisphere (Table 1). For the seven regions in Asia, Africa, Australia, South America, Europe, Antarctic, and Arctic defined in Xu et al. (2018), C-LSAT also have the largest stations number. The only exception happens in North America, where BE has the most

stations. But in BE, the stations from North American account for 85.7% of those from the Northern Hemisphere, which means that the stations from other parts of the Northern Hemisphere is only 14.3%. While in C-LSAT, stations from North America account for 30.7% of the Northern Hemisphere, and those from other parts of the Northern Hemisphere account for 69.3% (Figures 2a and 2b). Further, when the number of effective grid boxes in $5\,°\times5\,°$ grid containing observations are calculated between

1900 and 2017, we find that although Berkeley has more stations, C-LSAT has more effective grid boxes. In other words, although Berkeley's stations in the Northern Hemisphere is slightly higher than C-LSAT, the later has better data coverage in the whole Hemisphere (Figure 2c).

## 2.2 Sea surface temperature data

Currently, the following SST datasets are widely used in the corresponding community: HadSST3, ERSSTv5, Hadley Centre Sea Ice and Sea Surface Temperature dataset version 1 (HadISST1), and Centennial in situ Observation-Based Estimates of sea surface temperature version 2 (COBE2). The HadSST3 was derived from ICOADS R2.5 (1850-2006) and GTS (2007-present) observations (Kennedy et al., 2011). The ERSSTv5 dataset was developed by the NOAA NCEI, whose data sources

include ICOADS R3.0 SST data (including ships and buoys), near-surface Argo buoy data, and

HadISST2 sea ice data (Huang et al., 2017a). The HadISST1 was derived from the Met Office Marine

Data Bank (MDB), supplemented by the ICOADS SST data where the MDB data were missing. The

two-stage narrowed space optimization interpolation method was used in HadISST1 to obtain the sea

surface temperature dataset (Rayner et al., 2003). COBE2 was developed by the Japan Meteorological

Agency (JMA), using the original SST data from ICOADS R2.5 and sea ice concentration data

(Hirahara et al., 2014). A brief comparison between these datasets is shown in Table 2.

In general, only in situ observational data are used when merging LSAT and SST for the

commonly-used global ST datasets. For example, HadCRUT4 and BE use HadSST3 (the median of 100

ensemble datasets), whereas NOAAGlobalTemp and GISTEMP use ERSSTv5. Both HadSST3 and

ERSSTv5 use in situ data only. Other datasets, such as COBE and HadISST that use both in situ and

satellite data, are not used as a source in the merging of global ST data, although they are frequently

used in SST studies. Therefore, the HadSST3 and ERSSTv5 are selected and merged with the

C-LSAT1.3. The two other SST datasets with some satellite data previously merged (HadISST and

COBE2) are used for comparisons in this study.

## 3. Reconstruction of Global ST Dataset

### 3.1 Merging Schemes

As in other studies, the global ST dataset is merged with an LSAT and an SST dataset. In this study,



C-LSAT1.3 is merged with HadSST3 and ERSSTv5 separately. The final merged global ST dataset will be selected based on the comparison of the quality of the different merging schemes.

Before the merging, those two SST datasets are reprocessed. The median of the 100-member ensemble datasets in HadSST3 are calculated for each grid box (Kennedy et al., 2011). The ERSSTv5 has a value of -1.8 ℃ in many grid boxes in the Arctic and Southern Ocean, which refers to the areas where sea ice coverage is above 90%. Therefore, some special treatment is needed for these grid boxes. If the anomalies are 0 ℃ and SSTs are -1.8 ℃, then we replace these values of -1.8 ℃ in ERSSTv5 with missing values. The reference periods for both HadSST3 and ERSSTv5 are taken as 1961-1990.

The two merging schemes are described as follows:

(1) Merge1: C-LSAT1.3+HadSST3 (ensemble). Giving the resolution of both two datasets are 5 °×5 °, the two are directly merged using the ratios of ocean and land surface areas in a specific grid box.

(2) Merge2: C-LSAT1.3+ERSSTv5. Since the resolutions of the two are different, they are unified to the same resolution (1 °×1 ° resolution), and then merged using the ratios of ocean and land areas.

The merging process of C-LSAT1.3 and ERSST are described as follows:

(1) The anomalies are calculated in each grid boxes in reference to 1961-1990 base period for C-LSAT and ERSSTv5, respectively.

(2) For the ocean-land boundary part, the fraction of land and ocean areas is considered (see Figure



3, taking the January 2017 as an example). The detailed procedures are:

(a) Downscaling the land (C-LSAT1.3) and ocean data to $1°\times1°$ resolution. The resolution of the ocean data is $2°\times2°$, which is distributed in 4 grids of $1°\times1°$. The resolution of the land data is $5°\times5°$, which is distributed in 25 grids of $1°\times1°$.

(b) Using the ocean-land mask file to differentiate all grids in the world into land or ocean (download link: http://www.ncl.ucar.edu/Applications/Data/cdf/landsea.nc). The ocean-land mask file is based on Rand's global elevation and depth data, and the resolution of the ocean-land mask is modified to $1°\times1°$. The ocean-land mask file contains five types of markers: 0 for ocean, 1 for land, 2 for lakes, 3 for islands, and 4 for ice sheets. Marine data are used in parts of the ocean and ice sheets, and land data are used in parts of land, lakes, and small islands.

(c) The $1°\times1°$ ocean grid data and the $1°\times1°$ land grid data are spliced by the ocean-land mask to obtain $1°\times1°$ global ST grid data.

(d) The averaged surface temperature anomaly (STA) in each $5°\times5°$ grid is calculated as:

$$STA_{(5°\times5°)}(i,j) = \frac{1}{25}\left(\sum STA_{(1°\times1°)}(ii-2:ii+2, jj-2:jj+2)\right)$$

## 3.2 Comparison of two merged schemes

Based on the above methods, C-LSAT1.3 grid data is merged with HadSST3 and ERSSTv5 data to form the C-LSAT+HadSST (Merge1) and C-LSAT+ERSST (Merge2) global ST datasets, respectively.



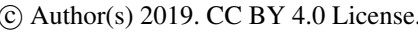

To choose a better merging scheme in CMST, Merge1 and Merge2 are compared in three aspects: spatial coverage and representativeness in high latitudes.

### 3.2.1 Global Coverage

The HadSST3 has not been interpolated, while the ERSSTv5 has been interpolated by EOTs (Huang et al., 2017). Because the data in interpolated boxes in ERSSTv5 are meaningful and the final dataset contains all the interpolated values, we do not distinguish whether boxes are interpolated or not and compare these boxes with HadSST3 directly in the following sections.

From Figure 4, we find that the spatial coverage of Merge2 increases steadily with time from Jan 1900 to Dec 2017. In contrast, in the early and middle 20th century, the coverage of Merge1 changed dramatically with time, and became steady and close to Merge2 after the late 20th century. It should be pointed out that if the ERSSTv5 original data are used (Merge2_obs), the coverage would be comparable with that of Merge1 in the whole period. Table3 also shows the global coverages of Merge1 and Merge2. The maximum coverage is found in Feb 1988 for Merge1 and in Jan 2000 for Merge2. The minimum coverage is found in Apr 1900 for Merge 1 and in Jun 1900 for Merge2. The mean coverage is calculated between 1900 and 2017. It can be seen from Table 3 that the Merge2 dataset has larger data coverage than Merge1 in all of the Coverage Mean, Coverage Max and Coverage Min. Although the difference between the two in Coverage Max is not very large, the difference in Coverage Means and Coverage Min between two merges is very large, which suggests that the coverage is mostly smaller in

Merge1 than Merge2. Therefore, although the original data coverage of HadSST3 and ERSSTv5 is similar with each other, but with the interpolation of EOTs, the later increased its coverage greatly, thus from the perspective of overall coverage, the dataset Merge2 is superior to Merge1. (Figure 4)

Furthermore, Figure 5 shows the spatial coverage of the average temperature anomalies per 20

5    years of Merge1 and Merge2. The six panels in Figures 5a and Figures 5b correspond to the 20-year mean temperature anomaly distribution over 1900-1919, 1920-1939, 1940-1959, 1960-1979, 1980-1999 and 2000-2017, respectively. In the early 20th century, it can be clearly seen that Merge1 lacked a large range of data in the equatorial region, the western region of the Southern Hemisphere and the high latitude zone of the Southern Hemisphere. In the mid-20th century, Merge1 lacked much data in the

10   high latitudes of the Southern Hemisphere. Merge1 remained lacking data at the high latitudes of the Southern Hemisphere by the end of the 20th century. In contrast, Merge2 exhibited data in global especially after 2000s. This is due to the rapid increase in the number of observations from Argo5obs (Argo floats between 0- and 5-m depth) between 2000 and 2006. Since 2006, the Argo5obs has maintained close to near-global coverage. In the high latitude region, the coverage of the Merge1 dataset

15   is also smaller than that of Merge2, which may critically impact the assessment of climate over the Arctic. This is mainly because the spatial coverage of ICOADS R3.0 used in Merge2 is slightly higher than R2.5 used in Merge1, especially in the south of 60°S and north of 60°N (Huang et al., 2017). Therefore, the coverage of the Merge1 is clearly lower than that of Merge2, particularly in the

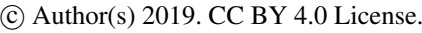

equatorial region and Southern Hemisphere. Therefore, with respect to the spatial coverage of each period, Merge2 has a much better spatial coverage, especially in the early 20th century.

### 3.3.2 Representativeness in high latitudes

To accurately compare the global and regional temperature changes between Merge1 and Merge2, we also introduce the COBE2 and HadISST1, which have satellite data integrated. First, the C-LSAT1.3 and COBE2, C-LSAT1.3 and HadISST1 datasets were merged in a similar way to form Merge3 (C-LSAT+COBE) and Merge4 (C-LSAT+HadISST) datasets. Second, the monthly temperature anomalies of Merge1-4 relatively to same baseline period (1961-1990) are calculated. The arithmetic mean of the four merged datasets was calculated for monthly temperature anomalies at each grid. As we know, each merging schemes would have uncertainties caused by different SST datasets, while the ensemble mean of all the merging datasets would have the least uncertainties. So the annual mean time series was calculated from the mean monthly temperature anomalies as a benchmark (reference series) for the two schemes.

From north to the south, the global ST is divided into five latitude zones: 90 ˚N-60 ˚N, 60 ˚N-30 ˚N, 30 ˚N-30 ˚S, 30 ˚S-60 ˚S, and 60 ˚S-90 ˚S. The reference series is subtracted from Merge2 and Merge1 datasets to obtain a difference series for each region. The comparison between the two schemes shows that the difference in mid-latitude and low latitude is small (figure omitted). The difference is large in the high latitudes (Figure 6). In 90 ˚N-60 ˚N, the difference between Merge2 and the reference series is

steadily close to the 0 line during 1900 - 2017, while the difference between Merge1 and the reference

series is colder at 1900s-1920s and warmer at 1930s-1980s and in the later 1990s. In 60 ̊S-90 ̊S, the

time series of Merge1 (1945) started later than Merge2 (1900), and the difference between Merge1 and

the reference series (blue) is abnormally large during 1945-1960s. While the difference between Merge2

and the reference series (red) is still very small. The large difference in Figure 6b may be associated

with small sampling size of the difference, or small coverage of Merge1, but the Merge2 agree well with

what we have expected.

The correlation coefficients between the time series of Merge1 and the reference series and

between Merge2 and the reference sequence in each latitude zone are calculated. The results show that

the correlation coefficients between Merge1 (Merge2) and the reference series are similar for the globe,

the Southern Hemisphere, the Northern Hemisphere and the mid-low latitudes, which exceed 0.98.

Compared with the reference series in the high latitude zone, Merge2 shows much more consistence

than Merge1. At 60 ̊S-90 ̊S, the correlation coefficient of Merge2 (0.90) is much larger than that of

Merge1 (0.30). At 90 ̊N-60 ̊N, the correlation coefficient of Merge2 (0.99) is slightly larger than that of

Merge1 (0.97).

In summary, compared with Merge1, Merge2 dataset is superior in terms of global coverage,

spatial distribution and the temporal change with the reference series. The possible reason is that the

ocean data used by the ERSSTv5 dataset are the latest ICOADS R3.0 data, whereas the ocean data used



by the HadSST3 dataset are from ICOADS R2.5. Also, the ERSSTv5 data incorporate more observations (such as Argo5obs). Based on the above analysis, Merge2 is used as the final scheme in the later sections, which is named CMST (China Merged Surface Temperature) in the following sections.

# 4. Comparison of CMST with other existing datasets

## 4.1 Spatial Coverage

Spatial coverages may differ among the following products because different spatial smoothing or interpolation method are applied: HadCRUT4.6.0.0 is a non-interpolated observation dataset. NOAAGlobalTemp v4 is first interpolated by EOTs in both LSAT and SST, and then masked based on actual observation availability. GISTEMP v3 250km-Smoothing (defined as GISTEMP1) is interpolated with a small scan radius. CMST is interpolated by EOTs in SST but no interpolation is applied in LSAT.

First, the monthly coverage is calculated by the ratio of the areas between valid grid boxes and total grid boxes in HadCURT4, NOAAGlobalTemp, CMST, and GISTEMP1 (Figure 7). Figure 7a shows that the area coverage in CMST is larger than those in other datasets in aspects of Coverage Max, Coverage Min, and Coverage Mean. Particularly the Coverage Min in CMST is much larger than those in the other datasets (Figure 7a). Second, the monthly coverage is averaged to obtain the annual average. It is shown that the coverage of CMST is larger than those of the other three datasets at any time (Figure 7b). Furthermore, the multi-year averaged coverage between 1900 and 2017 was calculated, which is 76,

58, 71, and 70%, respectively, in CMST, HadCRUT4, NOAAGlobalTemp, and GISTEMP1. In other words, the coverage in CMST is not only much larger than that in the dataset without interpolation such as HadCRUT4, but also larger than those in the dataset with interpolation such as GISTEMP1 and NOAAGlobalTemp.

5    The reasons why the coverage of CMST is greater than those of the other datasets are as follows: The spatial coverage of land data (CRUTEM4) in HadCRUT4 is smaller than that of C-LSAT in CMST (Xu et al., 2018), and the spatial coverage of marine data (HasSST3) in HadCRUT4 is also smaller than that of ERSSTv5 in CMST. The higher coverage of marine data results from two aspects: (a) The ocean data (ERSSTv5) used by CMST has additional sources of Argo data and uses ICOADS R3.0 containing more ship and buoy data. (b) The ocean data of HadCRUT4 has not been interpolated, while the ocean data used by CMST has been interpolated. The spatial coverage of the land dataset (GHCNm v3) in NOAAGlobalTemp is less than that of C-LSAT in CMST. The spatial coverage of the marine dataset (ERSSTv4) is also less than that in ERSSTv5, as ERSSTv5 incorporated new ICOADS data and added a decade of Argo floats data. Additionally, GISTEMP1 has the same land dataset as NOAAGlobalTemp so that its coverage is less than that in CMST, and its marine dataset is the same as that of CMST. Therefore, the spatial coverage of GISTEMP1 is less than that of CMST.

It should be noted that, the data coverage of GISTEMP1 increases rapidly during the 1950s (Figure 7b), which is mainly due to the rapid increase in Antarctic (60 °S-90 °S; Figure 8b). As in CMST, the

station data of GISTEMP1 in Antarctic is mostly from SCAR (Hansen et al., 2010). The differences

between these two datasets are that GISTEMP1 using the baseline period of 1951-1980 while CMST

using 1961-1990. So that GISTEMP1 reserved more short term stations in 1951-1980.

From Figure 7, we can see that HadCRUT4 and NOAAGlobalTemp have two minimum coverage

in around 1918 and 1943/1944. However, CMST and GISTEMP1 do not have these minima. Similar to

Section 3.3, we calculated the data coverage in five latitude zones and noticed that the data coverages of

HadCRUT4 and NOAAGlobalTemp have the greater fluctuations in the latitude zones of the 30°N-30°S

and 30°S-60°S. To find the latitude zone with the greatest impact on global coverage in 30°N-60°S, we

divided these latitude zones into 20°N-10°S, 10°N-20°S, 0°-30°S, 10°S-40°S, and 20°S-50°S. It is

found that the minimum value of 30°S-60°S coverage is the smallest, which has the greatest impact on

global coverage. Therefore, the reason for small spatial coverage of HadCRUT4 and

NOAAGlobalTemp is mainly due to the small coverage of the latitude zone of 30°S-60°S.

Since the 30°S-60°S latitude zone is dominated by oceans, the change of ST coverage in the

30°S-60°S latitude zone is likely related to the change of SST coverage. This is consistent with the

study by Vose et al. (2012), who noted that from the early twentieth century to the present, the coverage

of SST increased from 30% to 70% and the coverage of marine data decreased significantly during the

two World Wars. The decrease in the coverage of HadCRUT4 and NOAAGlobalTemp is very clear

during the period of the two World Wars. For CMST and GISTEMP, coverage is less affected during the

two World Wars because ERSSTv5 has been interpolated in many observation missing grid boxes.

## 4.2 Surface temperature trends

Li et al. (2019) shows that the recent global mean ST warming trend since 1998 derived from CMST increases slightly comparing with existing datasets, and is statistically significant. And it becomes closer among the newly developed global observational data (CMST), remote sensed/Buoy network infilled dataset, and adjusted reanalysis data (Cowtan and Way, 2014; Huang J et al., 2017; Simonds et al., 2017). Similar to Li et al. (2019), the temperature trends for the period of 1900-2017 in different latitudinal belts are compared among GISTEMP v3 250km-Smoothing (defined as GISTEMP1) , GISTEMP v3 1200km-Smoothing (defined as GISTEMP2), BEST with air temperatures over sea ice (defined as BEST1), BEST with water temperatures below sea ice (defined as BEST2), NOAAGlobalTemp, HadCRUT4, and CMST (Table 4).

Firstly, the ST trends in every region were compared. The temperature trend in the Northern Hemisphere high latitude is the largest (≥0.116°C/decade), and becomes lower in the mid-latitudes of the Northern Hemisphere, the mid-latitudes of the Southern Hemisphere, and the low latitudes. The lowest temperature trend is in the high latitudes of the Southern Hemisphere.

Secondly, the differences in the long-term trends of STs in different latitude zones are compared. The temperature trends with largest difference occur in the high latitudes. At the high latitudes of the Southern Hemisphere, temperature trend is the highest in HadCRUT4 (0.114±0.019 °C/decade), and is

the lowest in NOAAGlobalTemp v4 (0.031±0.011℃/decade). The largest difference between the

highest and the lowest temperature trends is 0.083℃/decade. In the high latitudes of the Northern

Hemisphere, temperature trend is the highest in GISTEMP2 (0.164±0.014℃/decade), and is the lowest

in CMST (0.116±0.012℃/decade). The maximum difference is 0.048℃/decade. In the middle and low

latitudes, the biggest difference is in the low latitude (0.018℃/decade).

Finally, the uncertainty range of the temperature trend of each dataset in different latitudes is

compared. The uncertainty of every dataset is very small in the middle and low latitudes, and the largest

in the high latitudes. In the high latitudes of the Southern Hemisphere, the uncertainty in CMST is the

smallest. In the Northern Hemisphere high latitudes, the uncertainty in CMST is larger than that of

BEST2 but smaller than other datasets.

## 4.3 Inter-annual variations

Figure 9 shows the time series of the global ST anomalies for the period 1990-2017 in seven

datasets, which are calculated by the area-weighted average. From 1900 to 2017, the temperature

anomalies show a clear warming trend. In the CMST, the highest temperature anomaly is 0.82℃ in

2016. There is a significant warming trend from the 1910s to the 1940s and from the 1960s to 2017. In

contrast, there is a trend of cooling from the 1940s to the 1950s. These changes are very consistent with

other datasets, and are related to the changes in El Niño and La Niña events, volcanic eruptions, sea ice

cover, and other factors (Simmons et al., 2017). Overall, the global ST changes in CMST and other



datasets are similar over 1900-2017. In the period over 1920s-1970s, CMST is slightly lower than other

datasets, whereas HadCRUT4 is slightly higher than other datasets. The maximum difference between

CMST and HadCRUT4 is in 1938 and 1948, and the difference in temperature anomalies in these two

years is 0.18 ℃. In 1938, the temperature anomalies are -0.17 ℃ and 0.01 ℃ in CMST and HadCRUT4,

respectively. In 1948, the temperature anomalies are -0.20 ℃ and -0.02 ℃ in CMST and CRUT4,

respectively.

The time series of ST anomalies in the seven datasets are also divided into the Northern

Hemisphere (a), the Southern Hemisphere (b), and five latitudinal zones in 90 ℃N-60 ℃N (c), 60 ℃N-30 ℃N

(e), 30 ℃N-30 ℃S (g), 30 ℃S-60 ℃S (f), and 60 ℃S-90 ℃S (h). Obviously (Figure 10a), the time series of

temperature anomalies in every dataset is very consistent in the Northern Hemisphere. At the low

latitudes (Figure. 10e, f, g), the maximum ST of several datasets occurs in 2016, whereas the minimum

occurs in different years. The minimum ST appears in 1917 in most datasets (GISTEMP1, GISTEMP2,

BEST1, BEST2, HadCRUT4, and CMST), but it appears in 1908, 1909, and 1910 in

NOAAGlobalTemp (Figure. 10a).

In the mid-latitude zone (Figure. 10e, f), the times with maximum ST in seven datasets are

generally consistent. The maximum ST occurs in 2015 in the 60 ℃N-30 ℃N and in 2017 in the 30 ℃S-60 ℃S.

The times with the minimum ST appears to be the same in seven datasets. The minimum ST appears in

1912 in the 60 ℃N-30 ℃N, and in 1911 in the 30 ℃S-60 ℃S. In the high latitudes of the Northern Hemisphere,



the maximum ST consistently occurs in 2016, and the minimum ST consistently occurs in 1902.

In the high latitudes of the Southern Hemisphere (Figure. 10d), the CMST is consistent with all the series derived from other datasets after 1960. There are many fewer stations/grid boxes in the Antarctic/higher latitudes and therefore larger variances before 1960.

## 5. Data Availability

The datasets used in CMST were derived from published data from the NHMS (China, Russia, USA, Canada, Australia, some Asian countries, etc.) or climate data research institutions (UK/CRU, NOAA/NCEI). Part of the data are exchanged from some countries or regions, and therefore will be conditionally available to public. Details of the data sources are as follows: The C-LSAT 1.3 in gridded form with a solution of $5\,°\mathrm{x}\,5\,°$ developed by SUN Yat-Sen University (SYSU) & China Meteorological Administration is available on the Climate Explorer website of the Royal Netherlands Meteorological Institute (KNMI) (http://climexp.knmi.nl/select.cgi?id=someone@somewhere&field=clsat_tavg). ERSST.v5 is from the US NOAA/NCEI at https://www.ncdc.noaa.gov/data-access/marineocean-data/extended-reconstructed-sea-surface-temperature-ersst-v5. The China Merged Surface Temperature Data (CMST) dataset developed by SYSU is currently public released on the Climate Explorer website of the Royal Netherlands Meteorological Institute (KNMI) (http://climexp.knmi.nl/select.cgi?id=someone@somewhere&field=cmst ). With the

digital object identifiers (DOIs) (https://doi.pangaea.de/10.1594/PANGAEA.901295) issued for the data

sets (Yun et al., 2019), we hope to have provided a repository of a new global ST analyses across the

past 120 yr from present back to year 1900, for the public as well as for the scientific user community.

## 6. Conclusion

Based on the LSAT dataset (C-LSAT1.3) and SST dataset (ERSSTv5), a new global ST dataset of

CMST (China Merged Surface Temperature) has been developed. This dataset was completed by the

cooperation between Sun Yat-sen University (SYSU), China Meteorological Administration (CMA),

and the United States NOAA/NCEI. In CMST, we found:

1) The spatial coverage is larger when C-LSAT1.3 and ERSSTv5 are merged, and is smaller when

C-LSAT1.3 and HadSST3 is merged, especially in the Polar Regions. And the former (CMST) is also

superior in terms of spatial distribution and the temporal change with the reference series (derived from

average of merges of C-LSAT1.3 and four SST datasets).

2) The LSAT in CMST uses a high-quality C-LSAT1.3. More than 6,000 stations are added to the

previous version of C-LSAT1.0 (Xu et al., 2018), which increase the data coverage. The newly added

stations are mainly from ISTI dataset. The SST in CMST uses ERSSTv5 that uses the ocean data from

the latest ICOADS R3.0 and incorporates multiple types of observations. Compared with other existing

global ST datasets, the CMST increases the overall coverage over global land and ocean surface.





3) The time series in CMST in the global and mid-low latitudes are overall consistent with the other merged datasets at both inter-annual and inter-decadal timescales. Therefore the temperature trend of CMST from 1900 to 2017 is consistent with those of the other datasets. In the high-latitude zones where the differences of temperature trend is usually large, the trend of CMST has the small uncertainty range,

which can enable us to capture the major climate changes in the high latitudes of the Northern and Southern Hemispheres.

**Acknowledgement**: This research is supported by National Key R&D Program of China (Grant: 2017YFC1502301), the Natural Science Foundation of China (Grant: 91546117), the Ministry of Science and Technology of China (Grant: GYHY201406016), and the China Postdoctoral Science Foundation (Grant: 2018M640848). We thank many

contributors who contribute to the establishment of this dataset.

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



Table1. Comparison of the station numbers of the LSAT dataset during 1900 - 2017 (data length greater than 15 years)

|  | C-LSAT | CRUTEM4 | GHCN | Berkeley |
|---|---|---|---|---|
| Global | **13687** | 9415 | 6871 | 12371 |
| Northern Hemisphere | 11270 | 7881 | 5633 | **11825** |
| Southern Hemisphere | **2418** | 1535 | 1238 | 548 |
| Africa | **922** | 749 | 586 | 367 |
| Asia | **2747** | 1831 | 1129 | 369 |
| Australia | **1022** | 388 | 563 | 91 |
| Europe | **3041** | 2177 | 930 | 334 |
| North America | 3462 | 2058 | 2699 | **10133** |
| South America | **753** | 669 | 340 | 261 |
| Arctic | **1105** | 1050 | 278 | 389 |
| Antarctic | **104** | 36 | 36 | 27 |





Table 2. Current international marine dataset for climate change research

| Datasets | Resolution | Time | Mainly used observation data | Satellite data |
|---|---|---|---|---|
| HadSST3 | 5 °×5 ° | 1850- | ICOADS R2.5 and some GTS data | No |
| ERSST.v5 | 2 °×2 ° | 1854- | ICOADS R3.0, Argo temperature above 5 m depth (Argo5obs), HadISST2 sea ice concentration, HadNMAT2, WOISST, Unadjusted SST | No |
| HadISST1 | 1 °×1 ° | 1870- | Met Office Marine Data Bank(MDB), GTS data (Since 1982), ICOADS (Use ICOADS SST data as a supplement in places where MDB data is missing) | Yes |
| COBE-SST2 | 1 °×1 ° | 1850- | ICOADS R2.5, MDB | Yes |





Table 3. Mean, max and min of monthly global coverage between 1900-01 and 2017-12

| Dataset | Coverage Mean | Coverage Max | Coverage Min |
|---------|---------------|--------------|--------------|
| Merge1  | 0.761         | 0.822        | 0.658        |
| Merge2  | 0.588         | 0.784        | 0.305        |

(c) Author(s) 2019. CC BY 4.0 License.



Table 4. Regional ST trends for different latitude zones from 1900 to 2017 ( ℃/decade)

|  | 90 ˚N – 60 ˚N | 60 ˚N – 30 ˚N | 30 ˚N – 30 ˚S | 30 ˚S – 60 ˚S | 60 ˚S – 90 ˚S |
|---|---|---|---|---|---|
| CMST | 0.116±0.012 | 0.098±0.006 | 0.082±0.005 | 0.080±0.003 | 0.046±0.004 |
| BEST1 | 0.149±0.016 | 0.104±0.006 | 0.071±0.005 | 0.090±0.003 | 0.113±0.008 |
| BEST2 | 0.118±0.010 | 0.102±0.006 | 0.071±0.005 | 0.086±0.003 | 0.055±0.005 |
| HadCRUT4 | 0.143±0.015 | 0.096±0.006 | 0.066±0.004 | 0.087±0.003 | 0.114±0.019 |
| GISTEMP1 | 0.142±0.013 | 0.090±0.006 | 0.077±0.004 | 0.085±0.002 | 0.037±0.006 |
| GISTEMP2 | 0.164±0.014 | 0.093±0.007 | 0.080±0.004 | 0.085±0.002 | 0.073±0.009 |
| NOAAGlobalTemp | 0.127±0.012 | 0.094±0.006 | 0.084±0.004 | 0.079±0.003 | 0.031±0.011 |



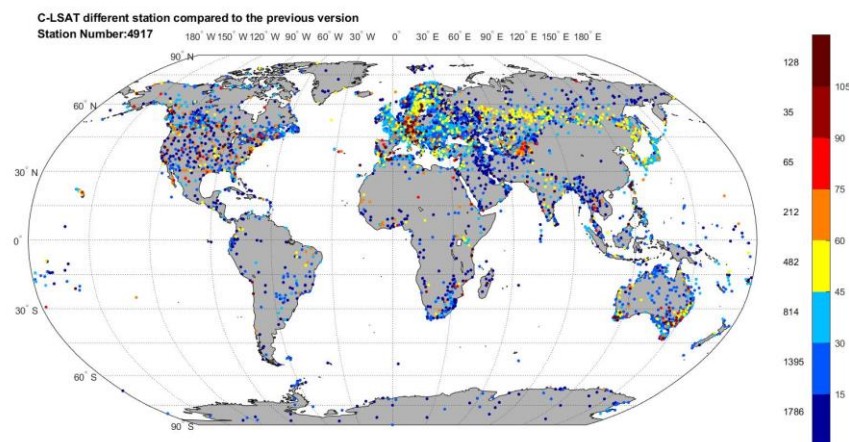

Figure 1. Stations added in C-LSAT version 1.3 between 1900 and 2017. The number on the right side of the color bar is the length of time and the number on the left side is the stations' number corresponding to a length of time.





(a)

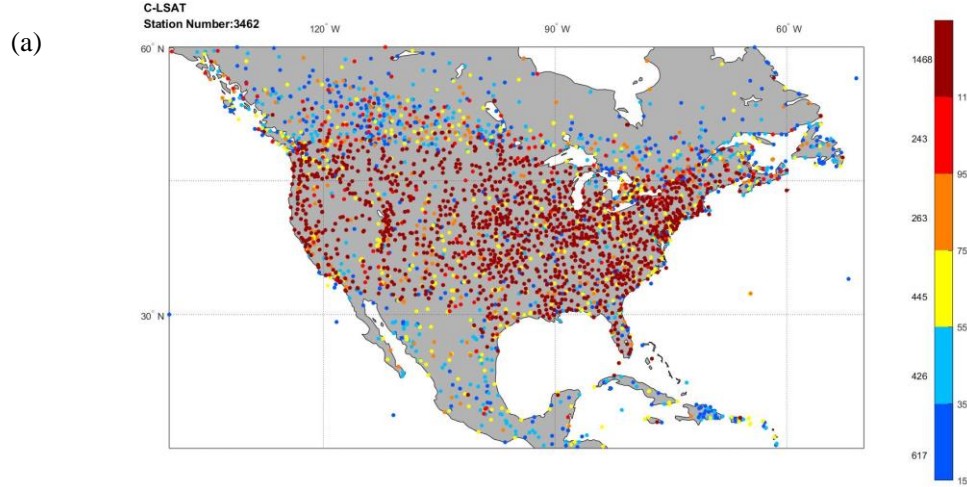

(b)

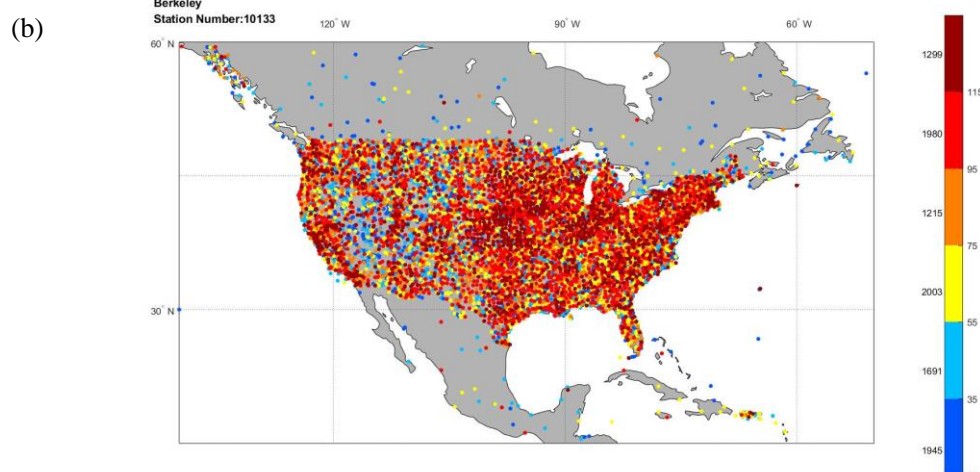



(c)

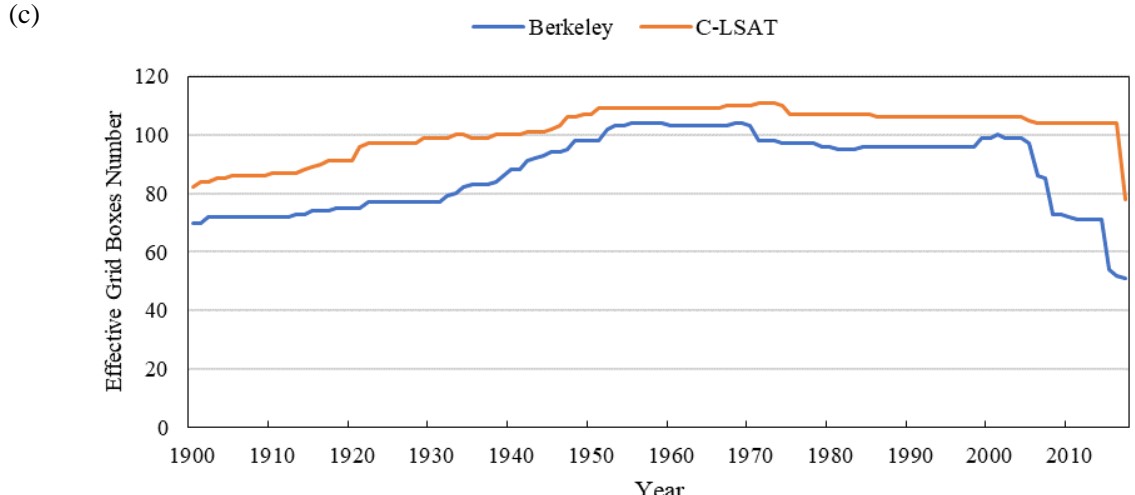

Figure 2. The distribution of the stations from 1900 to 2017 for C-LSAT (a) and BE (b), and the comparisons of effective grid box numbers in North America (c).





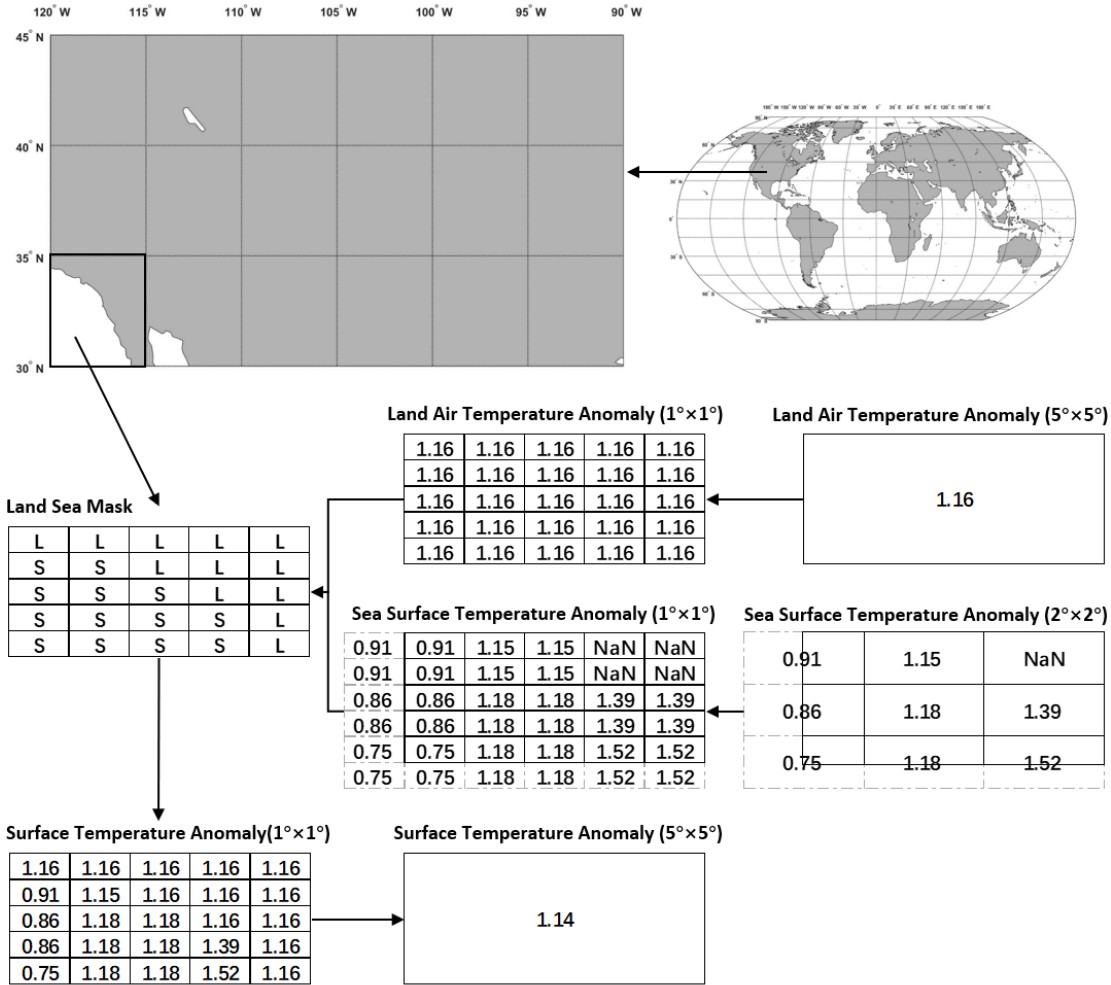

Figure 3. Calculation method for temperature anomalies with a resolution of 5 °×5 °in the grid containing ocean
and land



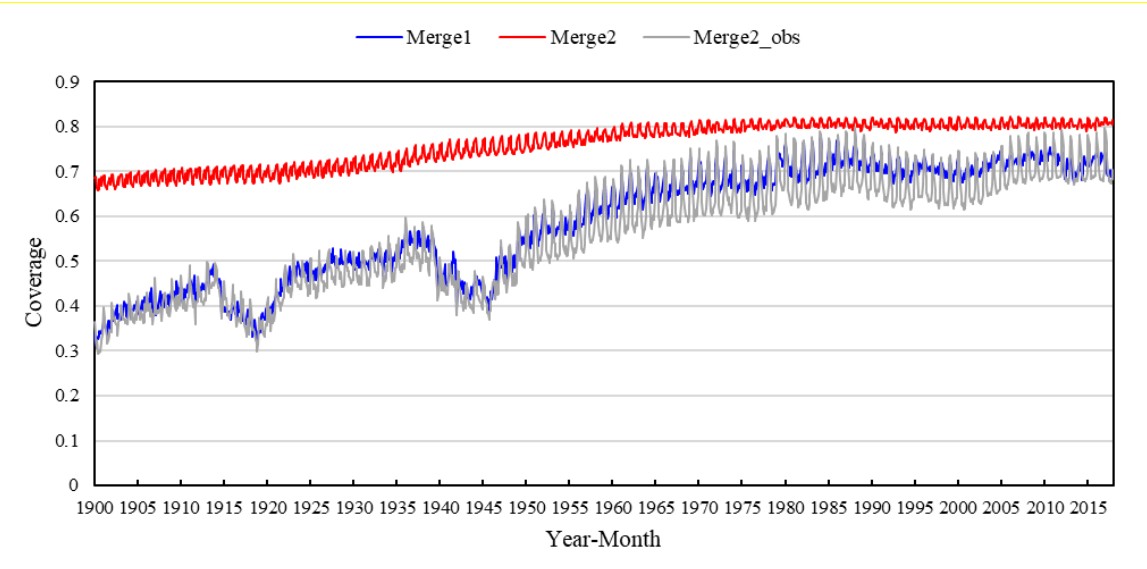

Figure 4. Comparison of monthly global coverage of the two datasets during 1900 to 2017. The grey curve shows Merge2 but with original data used.

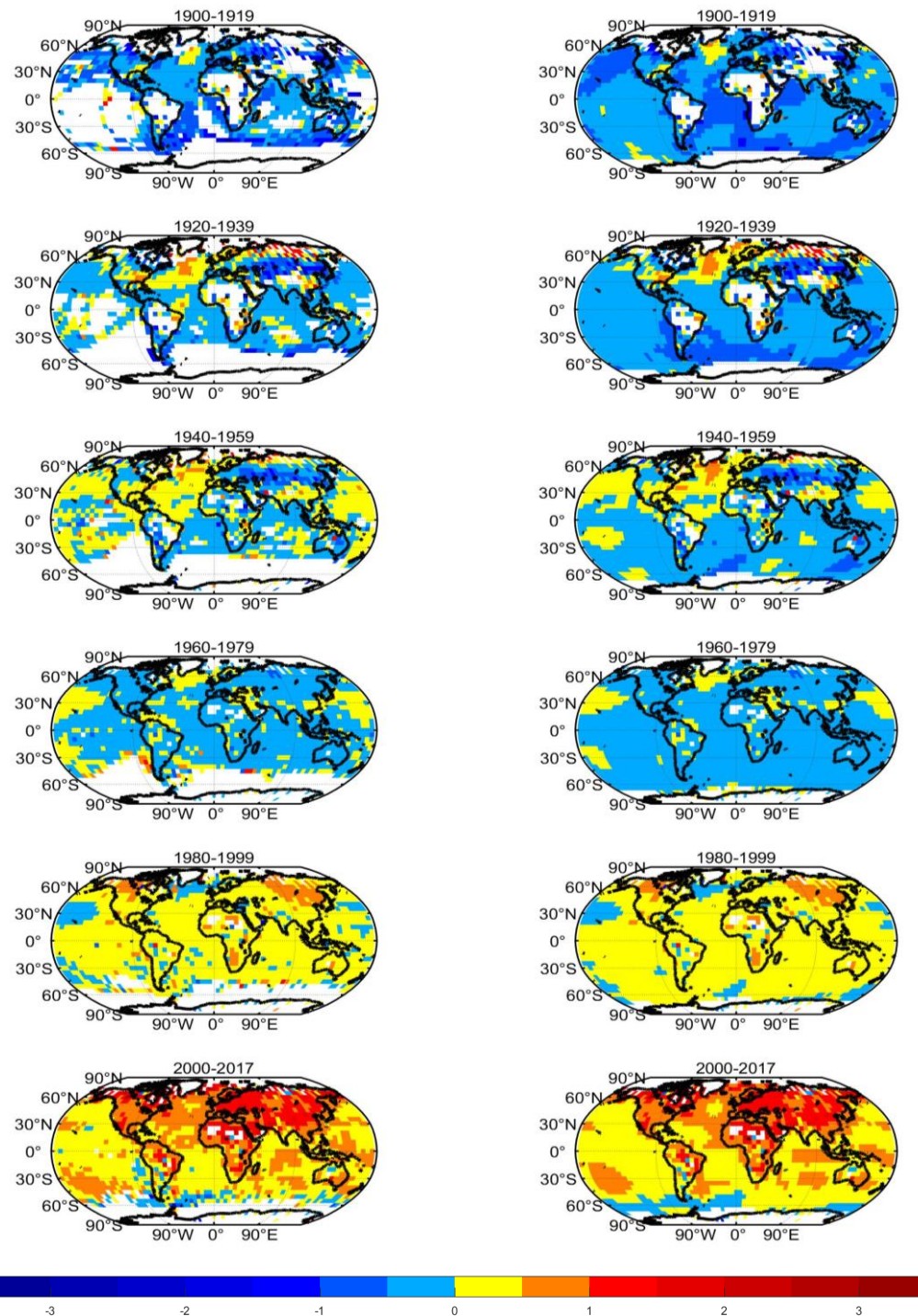

Figure 5. Spatial distribution of 20-year average temperature anomalies between 1900 and 2017 in Merge1 (left)
and Merge2 (right).



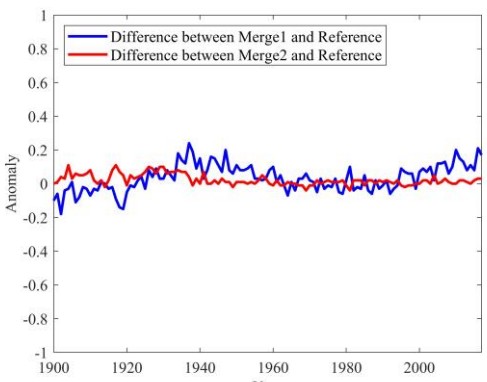 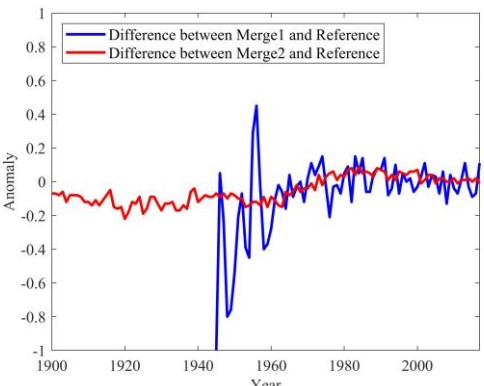

Figure 6. Differences between the merged series and the reference series during 1900 – 2017 in (left) 90 ̊N-60 ̊N and (right) 60 ̊S-90 ̊S. The difference between Merge1 and the reference series is blue, and the difference between Merge2 and the reference series is red.

(a)

(b)

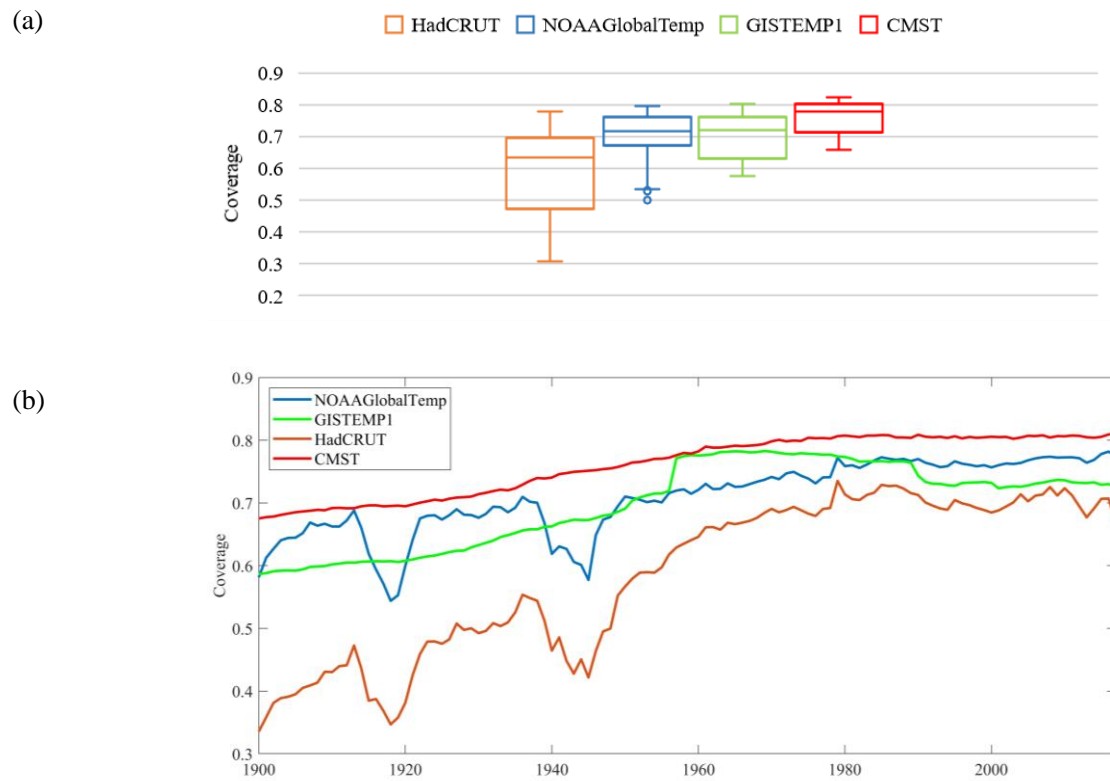

Figure 7. Comparison of global ST dataset coverage between 1900 and 2017 (a) monthly coverage for all grid boxes; (b) annual average of coverage of monthly grid data.




(a)

(b)

(c)

(d)

(e)

NOAAGlobalTemp    GISTEMP1    HadCRUT    CMST

Figure 8. Comparison of annual averages of coverage of ST datasets for each latitude zone between 1900 and 2017 in (a) 90 °N-60 °N, (b) 60 °S-90 °S, (c) 60 °N-30 °N, (d) 30 °S-60 °S and (e) 30 °N-30 °S).





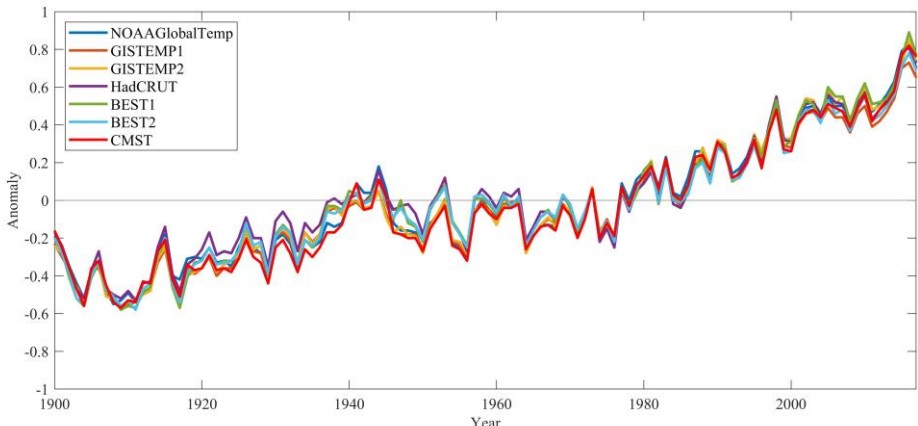

Figure 9. Comparison of global mean ST anomalies series during 1900-2017 by different datasets (relative to 1961-1990)



(a)

(b)

(c)

(d)

(e)

(f)





(g)

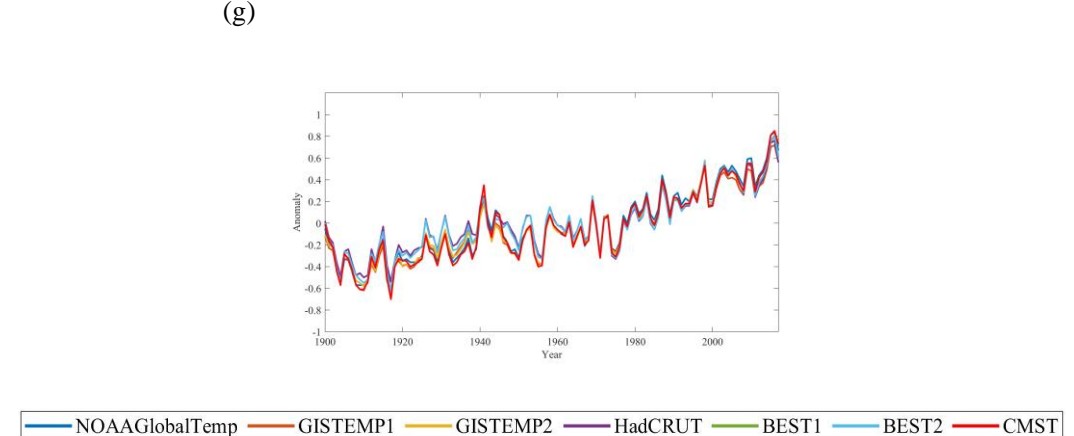

Figure 10. Comparison of regional ST anomalies series during 1900 – 2017 in (a) NH, (b) SH, (c) 90 ̊N-60 ̊N, (d) 60 ̊S-90 ̊S, (e) 60 ̊N-30 ̊N, (f) 30 ̊S-60 ̊S and (g) 30 ̊N-30 ̊S.