# Peer review of "A new merge of global surface temperature datasets since the start of the 20th Century"

_Earth System Science Data, 2019_

## Short Comment (SC1) · 19 Jun 2019

The characterization of Berkeley Earth station counts presented in this paper (e.g. Table 1) are grossly incorrect and in all cases too small.

GHCN and CRUTEM are sources used in the construction of the Berkeley Earth data set, so it would hardly be reasonable to expect the numbers to ever be smaller than GHCN.

A quick review suggests the following Berkeley Earth station counts (as of Feb 2019) for locations having at least 15 years of data during the interval 1900 to 2017:

Global 27564, Northern Hemisphere 24782, Southern Hemisphere 2780, Africa 983, Asia 4185, Australia 1029, Europe 3971, N. Amer. 14587, S. Amer. 862, Antarctic 48

I note that these numbers are often twice as large as reported in Table 1 and sometimes more than 10 times larger. I also suspect the characterization of Berkeley Earth presented in Figure 2 (b & c) is incorrect.

Consequently, the textual discussion about C-LSAT having the most stations is also generally incorrect.

I do not know why the figures reported in this paper regarding Berkeley Earth are so inaccurate. I do not believe that they are simply out-of-date as I do not believe that there was ever a time when the values reported in this paper would have been correct. Rather, it appears that there has been some problem with the handling or analysis of the data. We would be happy to assist the authors in identifying the source of the problem.

Sincerely,

Dr. Robert Rohde

Lead Scientist, Berkeley Earth

---

## Author Comment (AC1) · 19 Jun 2019

Dear Dr Robert Rohde,

This is a quick response. Thank you very much for pointing out this problem and offering help to us. Possible reason is that we have combined the stations with similar latitudes and longitudes into one sites. We will check more carefully on this, and update the information as soon as possible.

Regards

Qingxiang

2019.

---

## Author Comment (AC2) · 28 Jun 2019

Following is what we do according to your previous comments, Xiang Yun has send this to you, but maybe you did not get it. So I put them here. Please give some directions if there is something wrong.

Dear Dr Robert Rohde,

Thank you for pointing out the stations number problem for Berkeley Earth and offering help to us. I am not sure if this problem is due to a misunderstood way I get the station data (may be I find the wrong website to get station data), hope you can point it out. The Berkeley site data is accessed as follows: First, we open the website (http://www.berkeleyearth.dev/analysis-code), click on

the Download, find the file (site_complete_detail.txt) in the Download packet, get the station data. Then we open the site_complete_detail.txt file and import the site_complete_detail.txt file into site_complete_detail_no_illup.xlsx, we have 272556 stations in site_complete_detail_no_illup.xlsx. In site_complete_detail_no_illup.xlsx, we find that some stations' begin time or end time, longitude, latitude are missing, so we delete these stations, then we combine the stations with similar station ID, and get the data_year.xlsx file, which has 16694 stations. Finally, stations with a time length of more than 15 years, latitude of -90∼90 and longitude of -180∼180 are selected, and the file of lat-lon-start-end-length.xlsx are obtained, with a total of 12,371 stations. Eight regions are defined, following Jones and Moberg (2003), for the seven continents of the world (Asia(5-60°N, 60-180°E), Africa(35°N-40°S,20°W-45°E), South America(15°N-55°S,30-80°W), Europe(35-60°N,15°W-60°E), North America(15-60°N,140-50°W), Australia(10-50°S,110-155°E), and Antarctic(60-90°N)) plus the Arctic(60-90°S), the station numbers of 8 regions are shown in table 1. Sincerely looking forward to your reply. Xiang Yun

———————————————————————

---

## Referee Comment (RC1) · Anonymous Referee #1 · 29 Jun 2019

This manuscript devoted to generating a new merge of global surface temperature anomaly dataset, through merging C-LSAT and ERSSTv5 datasets. However, there are several problems in the manuscript.

1. The global monthly temperature anomaly product of 5 degree seems to be too coarse to serve for investigation of temperature variation, especially for some regions such as the land and ocean convergence zones. Currently, development of climatic dataset should provide data of higher spatial resolution.

2. In Page 12 and Figure 3, To calculate surface temperature anomaly at 5-degree scale, the authors adopted averaged value of temperature anomaly at 1-degree scale. Moreover, spatial transformation of sea surface temperature anomaly from 2 to 1 degree is not robust. Some geographic statistic methods should be employed to guide

these spatial transformations of geographic data, such as Nearest Neighbor, Bilinear, Kriging, and Inverse Distance Weighting interpolations.

3. I noticed that the dataset was stored as txt files. It is better to give product in NetCDF format to provide the longitude and latitude information. In addition, in the "readme.txt", authors stated that -999.99 was set as the missing value; however, I noticed that 999.99 and NaN were used as the missing value in the specific data files.

4. It is necessary to reorganize the statement of each part and find a professional English Editor to improve the language quality of the ms, because the whole ms is rough to be read.

---

## Author Comment (AC3) · 5 Aug 2019

Dear Reviewer,

Thank you very much for your comments and suggestions which are very helpful to improve the manuscript. We provided a very detailed response to each of your comments, and revised the manuscript based on all the comments from you. We also provided a revised manuscript in tracked changes mode.

Best wishes,

Qingxiang

Response to your comments/suggestions:

[Figure]

This manuscript devoted to generating a new merge of global surface temperature anomaly dataset, through merging C-LSAT and ERSSTv5 datasets. However, there are several problems in the manuscript. 1. The global monthly temperature anomaly product of 5 degree seems to be too coarse to serve for investigation of temperature variation, especially for some regions such as the land and ocean convergence zones. Currently, development of climatic dataset should provide data of higher spatial resolution. Response: Thank you very much for the suggestions. This dataset - CMST, is still a global surface temperature dataset used to analyze the surface Temperature change in large scales since the start of 20 century. In this perspective, 5 degree would be enough for the large scale (global/hemispheric/continental) mean ST change due to the good representativeness of surface temperature. The similar global ST datasets like HadCRUT, NOAAGlobalTemp, GISTem are all currently in 5 degree resolution. As you mentioned, higher resolution datasets are very important for regional/local surface temperature variation analysis. In fact, we are developing the higher resolution global land surface air temperature dataset with advanced interpolation methods. But it is also very difficult to develop a high resolution dataset in good quality. Many factors like location, elevation, coastal proximity, topographic facet orientation, vertical atmospheric layer, topographic position, and orographic effectiveness of the terrain should be considered, especially in the construction of the climatology. (Daly et al, 2008). There are some high resolution global land surface air temperature datasets: CRU T4.03 (0.5*0.5), Berkerly Earth (1*1) (Rohde et al., 2013), and WorldClim (only climatology). CRU T4.03 (New et al., 2002) and WorldClim (Fick and Hijmans, 2017) is based on AUSPLINE (climatology) +IDW (anormaly), BE is based on Kriging. But we need to point out that the high resolution data need to pay more attention to the spatial distribution (lower RMSE, AME) rather than the temporal changes, which may lead some inhomogeneities in the grid series and thus affect the detection of the long-term trends. So this would be a different concern in some degree, we will discuss this in the future studies.

Ref: Daly, C., Halbleib, M., Smith, J.I., Gibson, W.P., Doggett, M.K., Taylor, G.H.,

Curtis, J., and Pasteris, P.A. 2008. Physiographically-sensitive mapping of temperature and precipitation across the conterminous United States. International Journal of Climatology, 28: 2031-2064. New M, Lister D, Hulme M, Makin I. 2002. A high-resolution data set of surface climate over global land areas. Climate Res. 21: 1–25. Fick S. E., R. J. Hijmans, 2017, WorldClim 2: new 1‐km spatial resolution climate surfaces for global land areas. International Journal of Climatology, 37: 4302–4315. https://doi.org/10.1002/joc.5086 Rohde R., R. A. Muller, et al. (2013) Berkeley Earth Temperature Averaging Process. Geoinfor Geostat: An Overview 1:2. doi:10.4172/gigs.1000103

2. In Page 12 and Figure 3, to calculate surface temperature anomaly at 5-degree scale, the authors adopted averaged value of temperature anomaly at 1-degree scale. Moreover, spatial transformation of sea surface temperature anomaly from 2 to 1 degree is not robust. Some geographic statistic methods should be employed to guide these spatial transformations of geographic data, such as Nearest Neighbor, Bilinear, Kriging, and Inverse Distance Weighting interpolations. Response: Thank you very much for the suggestions. You are right when one does the similar downscaling in climatology (climatic normal) field. Due to the spatial inhomogeneity of sea surface temperature (SST), there will be a certain difference when different methods are used. But it will be a little different when downscaling the temperature anomalies (SSTA/SATA). The anomalies always have much better spatial representation especially in SSTA. Here in this manuscript, the purpose of downscaling the SSTA to 1*1 resolution is only for the convenience of merging the land and marine data. We downscaled the 2*2 sea surface temperature anomaly to the 1*1 resolution to match with the land surface air temperature data (C-LSAT) in 1*1 resolution (downscaled in the same method from 5 degree grid box). The slight difference between the simple averaged values with the interpolated values with geographic statistic methods will not matter much in the final CMST dataset. Based on above, we only simple adopt the average values for the SST anomalies downscaling.
3. I noticed that the dataset was stored as txt files. It is better to give product in NetCDF format to provide the longitude and latitude information. In addition, in the "readme.txt", authors stated that -999.99 was set as the missing value; however, I noticed that 999.99 and NaN were used as the missing value in the specific data files. Response: Thank you very much for the suggestions. We have transferred the ASCII code data into the NetCDF and updated the readme.txt. We will update them on the website whenever we can.

4. It is necessary to reorganize the statement of each part and find a professional English Editor to improve the language quality of the ms, because the whole ms is rough to be read. Response: Thank you very much for the suggestions. We have revised the ms as much as we can by the coauthors and some friends from English speaking areas, and will have a professional English Editor to improve it when the review process finished.

Please also note the supplement to this comment:
https://www.earth-syst-sci-data-discuss.net/essd-2019-80/essd-2019-80-AC3-supplement.pdf

**Supplement:**

**A new merge of global surface temperature datasets since the start of the 20th Century**

Xiang Yun[1,2], Boyin Huang [3], Jiayi Cheng[1#], Weihui Xu[4], Shaobo Qiao[1#], Qingxiang Li [1,#*]

1 School of Atmospheric Sciences and Guangdong Province Key Laboratory for Climate Change and Natural Disasters, SUN Yat-Sen University, Guangzhou, China

2 Chinese Academy of Meteorological Sciences, CMA, Beijing, China

3 National Centers of Environmental Information, NOAA, Asheville, USA

4 National Meteorological Information Center, CMA, Beijing, China

**Southern Laboratory of Ocean Science and Engineering (Guangdong Zhuhai), Zhuhai, China**

*Corresponding to: Qingxiang Li (liqingx5@mail.sysu.edu.cn)

**Abstract**

Global surface temperature (ST) datasets are the foundation for global climate change research. There are sSeveral global ST datasets have been developed by different groups in NOAA/NCEI, NASA/GISS and UKMO Hadley Centre & UEA/CRU. In this study,This study presents a new global ST dataset was presented, the named China Merged Surface Temperature (CMST) dataset. CMST is created by merging the China-Land Surface Air Temperature (C-LSAT1.3) with the sea surface temperature (SST) data from the Extended Reconstructed Sea Surface Temperature version 5 (ERSSTv5). The merge of C-LSAT and ERSSTv5 shows a high spatial coverage extended to the high latitudes and is more consistent with a reference of multi-datasets average in the Polar Regions.

Comparisons indicated that CMST is consistent with other existing global ST datasets in interannual-decadal variations and long-term trends at global, hemispheric, and regional scales from 1900 to 2017.  The CMST dataset can be used for global climate change assessment, monitoring, and detection. The CMST dataset presented here is publicly available

5   at: https://doi.pangaea.de/10.1594/PANGAEA.901295 (Yun et al., 2019) and has been published on the Climate Explorer website of the Royal Netherlands Meteorological Institute (KNMI) at: http://climexp.knmi.nl/select.cgi?id=someone@somewhere&field=cmst.

**1. Introduction**

The long-term trend of the global mean surface temperatures (GMST) is one of a common measure in observing the change of climate.  Therefore, the biases of the observed surface temperature (ST) dataset, particularly  the sampling bias of high latitudes stations , has received much attention in the past few years (Cowtan and Way, 2014; Jones, 2016; Li et al., 2017; Simonds et al., 2017; Huang et al., 2017a).  The optimization and improvement of  observed climate data is a long-term task, as a reference base for climate change research and verification benchmark for other climatic data products.

A total of four global land surface air temperature (LSAT) observation series and three global ST series were exhibited by The Intergovernmental Panel on Climate Change (IPCC, 2013) few years ago. . The four LSATs including the Climatic Research Unit (CRU) land surface air temperature, version 4 (CRUTEM4; Jones et al., 2012), Global Historical Climatology Network-monthly (GHCNm) temperature, version 3 (GHCNm v3; Lawrimore et al., 2011), Goddard Institute for Space Studies analysis of land surface air temperature (GISS; Hansen et al., 2010), and Berkeley Earth Surface

Temperature group land temperature (Berkeley; Rohde et al., 2013). he three global ST series are

the Met Office Hadley Centre and Climatic Research Unit Temperature version 4 (HadCRUT4; Morice

et al., 2012), Merged Land–Ocean Surface Temperature (MLOST; Vose et al., 2012b), and Goddard

Institute for Space Studies Surface Temperature Analysis (GISTEMP; Hansen et al., 2010).

5       All these datasets  indicated that the Earth has experienced a "warming hiatus" period  from

1998 to 2012.and this issue has been attracted many attention from the researchers

around the world. However, by analyzing the sea surface temperature (SST) and global ST  from the

National Oceanic and Atmospheric Administration / National Centers for Environmental Information

(NOAA/NCEI), Karl et al. (2015) suggested that the "warming hiatus" is due to the artifact of the data

10      processing. Besides that, Lewandowsky et al. (2015) noted that this short-term warming trend

"hiatus" is a conditional statistical artifact and it is not a real scientific fact. After correcting the

sampling biases of the temperature data  over the Arctic region from  few studies

 by using reanalysis data (Simonds et al., 2017), satellite remote sensing data (Cowtan

and Way, 2014), and Arctic buoy data (Wang et al., 2017), their results have come to a similar

15     conclusion.

These global ST data products have been updated over the past few years since the

publication of IPCC (2013). For instance, the NOAA has updated the Extended Reconstructed Sea

Surface Temperature (ERSST) version 3 to ERSSTv4 (Huang et al., 2015) and ERSSTv5 (Huang et al.,

2017), updated LSAT dataset GHCNm v3 to GHCNm v4 (Menne et al., 2018), and renamed MLOST to

NOAA Global Surface Temperature (NOAAGlobalTemp). The GISTEMP has been updated its SST

component to ERSSTv5 (Huang et al., 2017b). While CRUTEM has been updated to CRUTEM4.6. The

Met Office has updated the Hadley Centre SST to version 3 (HadSST3) using the median of 100

5    ensemble members. And lastly, tThe Berkeley team uses used the median of the HadSST3 ensembles of

HadSST3 to form the Berkeley Earth Surface Temperature (BEST) dataset.

The products' updates of these products are based on the advanced knowledge of data analysis

methodology or improved data availability. In general, the GMST has continuously been improved by

the increased number and area coverage of observational data over the land (LSAT) and in the oceans

10   (SST). There are two aspects to improve the LSAT datasets: . The first Firstly, is to increase the density

of stations and data coverage especially in the key areas with sparse observations. For example, the

number of observations is increased in both C-LSAT (Xu et al., 2018) and GHCNm v4 (Menne et al.,

2018) using a newly released International Surface Temperature Initiative (ISTI) datasets (Thorne et al.,

2011) or datasets through regional cooperation with Asian countries such as Vietnam and South Korea.

15   Coverage of datasets increases with larger number of observations and hence reducing the sampling

biases,The larger number of observations increases the coverage of datasets and therefore reduces the

sampling biases,  especially particulary at for high latitudes area (Polar Regionsregion) and in the

observation-sparse regions (such as South America and Africa). Next,  The the second aspect is to

improve improving the accuracy of regional climate changes. For example, the latest C-LSAT (Xu et al., 2018) has integrated more regional homogenization results, especially in over the China (Li et al., 2009; Xu et al., 2013), East Asia, Europe, Australia (Trevin, 2013), and Canada (Vincent et al., 2012).

On the other hand Similarly, there are also two aspects to improve the SST datasets: (1) integration of much better raw observational data and; (2) replacing a single analysis to multi-member ensemble analyses. For instances, ERSSTv5 is using the most recently available International Comprehensive Ocean-Atmosphere Data Set release 3.0 (ICOADS R3.0; Freeman et al., 2017), optimized climate modes and more accurate buoy data in adjusting the ship data. Meanwhile, HadSST3 introduces a variety of bias correction models and median SST of the 100 ensemble members was used as the best estimation.

The first is to integrate better raw observational data. For example, the ERSSTv5 uses the most recently available International Comprehensive Ocean-Atmosphere Data Set release 3.0 (ICOADS R3.0; Freeman et al., 2017), uses more accurate buoy data to adjust ship data, and uses optimized climate modes. The second is to replace a single analysis with multi-member ensemble analyses. For example, HadSST3 introduces a variety of bias correction models and uses the median SST from the 100 ensemble members as the best estimate.

Among all the existing global ST datasets (e.g.,for example, HadCRUT and NOAAGlobalTemp), the merging methods on in combining the land and ocean datasets are basically is very similar to each

other. The merging process of HadCRUT includes: First, the land and ocean data  were processed into 100 ensemble datasets according to the bias evaluation parameters,  in which the anomaly values  calculated separately for each grid box . Then, anomalies in the grid boxes of the land and ocean boundary  weighted by the fraction of land and ocean areas. If the land area covers less than 25 %, it is calculated as 25 %. If there is a measured SAT anomaly in a grid box covered with sea-ice, The SAT will be used to represent the SST anomaly. The HadCRUT ensemble dataset has a reference period  range from 1961- to 1990  with a resolution of $5°×5°$ (Morice et al., 2012). The merging process in NOAAGlobalTemp.  The first step is to identify  the LSAT/  SST low frequency changes by calculating the moving average of temperature anomaly data, followed by  the identification of the LSAT  SST residual high frequency changes via the Empirical Orthogonal Teleconnection (EOT) modes. Then, the low- and high-frequency components  were integrated together. Finally, average the SST data at $2°×2°$ resolution  into the grid at $5°×5°$ resolution, and the land and ocean reconstructions  were then merged into a global reconstruction similar to HadCRUT (Vose et al., 2012).

This study presents a new merged global ST dataset based on the recently developed C-LSAT and the latest ERSSTv5 using a method which is similar to the HadCRUT and NOAAGlobalTemp,  providing a new reference to the climate or  climate change studies. The remainder of this paper is arranged into different sections as below : The land and ocean datasets and their

updates are briefly introduced in section 2.The merging process of CMST is given in section 3.Section 4 discussed the comparisons of CMST with other existing ST datasets.The availability of the resulting dataset (Yun et al., 2019) is reported in section 5 and  summary of results are presented in section 6.

**2. Updates of land and ocean data**

**2.1 Land surface air temperature data**

The C-LSAT1.0 dataset (Xu et al., 2018) processed the SAT data since 1900 from a total of 14 data sources, these including three global data sources (CRUTEM 4.6, GHCNv3, and BEST), three regional data sources from Scientific Committee on Antarctic Research (SCAR),  daily dataset for European Climate Assessment (ECA&D), and Historical Instrumental climatological Surface Time series of the greater Alpine region (HISTALP), and eight national data sources from China, America, Russia, Canada, Australia, Korea, Japan, and Vietnam. Two steps were taken to ensure the homogeneity of the station time series: (1) the data series from the existing national homogenized datasets were directly integrated into C-LSAT without any change, which  is approximately 50 % of the stations in C-LSAT. (2) the inhomogeneities in the rest of the station series were detected and adjusted with the penalized maximal t-test method (Wang et al., 2007).

The C-LSAT version 1.3 dataset is used in this study. Compared  to the C-LSAT version 1.0

range from 1900 to 2014 (Xu et al., 2018), the version 1.3 is updated to December 2017. According to Xu et al. (2018), national, regional and global datasets are ranked as higher, middle and lower priorities, respectively. Based on the priority of the data resources, a total of 4917 high priority stations were added, while total of 1364 low priority stations were deleted. Most of the newly added raw data were obtained from the International Surface Temperature Initiative (ISTI) Projects and have been homogenized through the same approach as Xu et al. (2018). The distribution of these extra 3553 stations shown in Figure 1. According to Xu et al. (2018), the C-LSAT version 1.0 had some advantages over the existing global LSAT datasets in station number and spatial coverage. Thus, the current C-LSAT version 1.3 has more station number than the existing datasets in many regions over the global land surface. Figure 1 shows the extra stations compared to version 1.0 and Table 1 shows the comparison of the station number for different datasets, indicating an enhanced coverage and distribution/sampling of LSAT observations.

For the comparison purposes, other LSAT datasets were collected from CRUTEM4 (https://www.metoffice.gov.uk/hadobs/), GHCNm v3 (ftp://ftp.ncdc.noaa.gov/pub/) and Berkeley Earth (BE) (http://www.berkeleyearth.dev/). All these datasets above were downloaded in August 2018. The following calculations are based on stations with a time length greater than 15 years in between 1900 and 2017.

From a global and hemispheric perspective, the C-LSAT version 1.3 dataset has more stations than the other datasets in the Global and Southern Hemisphere (Table 1). Besides that, C-LSAT also have the largest stations number  for  seven regions - in Asia, Africa, Australia, South America, Europe, Antarctic, and Arctic as defined in Xu et al. (2018) The only exception  was in North America, where BE has the most stations number.  However,  for BE dataset,  stations from North American account for 85.7 % of those from the Northern Hemisphere, this meaning that the stations from other parts of the Northern Hemisphere is only 14.3 %. While  for C-LSAT dataset, stations from North America account for 30.7 % of the Northern Hemisphere, and those from other parts of the Northern Hemisphere account for 69.3 % (Figures 2a and 2b). Furthermore, when the number of effective grid boxes in $5° × 5°$ grid containing observations are calculated between 1900 and 2017, we  noticed that C-LSAT has more effective grid boxes even though  the Berkeley dataset has more stations number. In other words, although the Berkeley's stations number in the Northern Hemisphere is slightly higher than C-LSAT, the later one has better data coverage in the whole Hemisphere (Figure 2c).

**2.2 Sea surface temperature data**

Currently, the following SST datasets are widely used in the corresponding community: HadSST3, ERSSTv5, Hadley Centre Sea Ice and Sea Surface Temperature dataset version 1 (HadISST1), and Centennial in situ Observation-Based Estimates of sea surface temperature version 2 (COBE2). The

HadSST3 was derived from ICOADS R2.5 (1850-2006) and GTS (2007-present) observations (Kennedy et al., 2011). The ERSSTv5 dataset was developed by the NOAA NCEI,  where their data sources include ICOADS R3.0 SST data (including ships and buoys), near-surface Argo buoy data, and HadISST2 sea ice data (Huang et al., 2017a). The HadISST1 was derived from the Met Office Marine Data Bank (MDB), supplemented by the ICOADS SST data where the MDB data were missing. The two-stage narrowed space optimization interpolation method was used in HadISST1 to obtain the sea surface temperature dataset (Rayner et al., 2003). COBE2 was developed by the Japan Meteorological Agency (JMA), using the original SST data from ICOADS R2.5 and sea ice concentration data (Hirahara et al., 2014). A brief comparison between these datasets  shown in Table 2.

In general, only  in-situ observational data  were used when merging LSAT and SST for the commonly-used global ST datasets. For example, HadCRUT4 and BE use HadSST3 (the median of 100 ensemble datasets),  meanwhile the NOAAGlobalTemp and GISTEMP were using ERSSTv5. Both HadSST3 and ERSSTv5 datasets use only  in-situ data . Other datasets, such as COBE and HadISST  which is using both  in-situ and satellite data,  were not used as a source in the merging of global ST data, although they are frequently used in SST studies. Therefore, the HadSST3 and ERSSTv5 datasets  were selected and merged with the C-LSAT1.3. While other two SST datasets with some satellite data previously merged (HadISST and COBE2)  were

used for comparisons in this study.

**3. Reconstruction of Global ST Dataset**

**3.1 Merging Schemes**

Generally in previous studies, the global ST dataset  was merged with  LSAT and an SST dataset. In this study, C-LSAT1.3 is merged with HadSST3 and ERSSTv5 separately. The final merged global ST dataset will be selected based on the comparison of the quality of  different merging schemes. These two SST datasets are reprocessed before the merging.  The median of the 100-member ensemble datasets in HadSST3  were calculated for each grid box (Kennedy et al., 2011). The ERSSTv5 has a value of -1.8 ℃ in many grid boxes in the Arctic and Southern Ocean, which refers to the area where the sea ice coverage is above 90 %. Therefore, some special treatment is needed for these grid boxes. If the anomalies are 0 ℃ and SSTs are -1.8 ℃, then the value of -1.8 ℃ in ERSSTv5 will be replace with missing values.  The reference period for both HadSST3 and ERSSTv5  were taken as 1961-1990.

The two merging schemes are described as follows:

(1) Merge1: C-LSAT1.3+HadSST3 (ensemble). Giving the resolution of both two datasets are 5 °× 5 °, the two datasets  were directly merged using the ratios of ocean and land surface areas in a

specific grid box.

(2) Merge2: C-LSAT1.3+ERSSTv5. Since the resolution of the two datasets are different, they  were unified onto the same resolution ($1° \times 1°$ resolution), and then merged using the ratios of ocean and land areas.

The merging process of C-LSAT1.3 and ERSST are described as follows:

(1) The anomalies  were calculated in each grid boxes  with respect to  the base period 1961-1990  for C-LSAT and ERSSTv5, respectively.

(2) For the ocean-land boundary part, the fraction of land and ocean areas is considered (see Figure 3, taking the January 2017 as an example). The detailed procedures are:

(a)  Downscale the land (C-LSAT1.3) and ocean data to $1° \times 1°$ resolution. The resolution of the ocean data is $2° \times 2°$, which is distributed in 4 grids of $1° \times 1°$. The resolution of the land data is $5° \times 5°$, which is distributed in 25 grids of $1° \times 1°$.

(b) Using the ocean-land mask file to differentiate all grids globally into land or ocean (download link: http://www.ncl.ucar.edu/Applications/Data/cdf/landsea.nc). The ocean-land mask file is based on Rand's global elevation and depth data, and the resolution of the ocean-land mask is  re-gridded to $1° \times 1°$. The ocean-land mask file contains five types of markers: 0 for ocean, 1 for land, 2 for lakes, 3 for islands, and 4 for ice sheets. Marine data  was used in parts of the ocean and ice sheets, and land data  was used in parts of land, lakes, and small islands.

(c) The $1°\times1°$ ocean grid data and the $1°\times1°$ land grid data  were spliced by the ocean-land mask to obtain $1°\times1°$ global ST grid data.

(d) The averaged surface temperature anomaly (STA) in each $5°\times5°$ grid  was calculated as:

$$STA_{(5°\times5°)}(i,j) = \frac{1}{25}\left(\sum STA_{(1°\times1°)}(ii-2:ii+2, jj-2:jj+2)\right)$$

**3.2 Comparison of two merged schemes**

Based on the  methods above, C-LSAT1.3 grid data is merging with HadSST3 and ERSSTv5 data to form the C-LSAT+HadSST (Merge1) and C-LSAT+ERSST (Merge2) global ST datasets, respectively. In order to choose a better merging scheme in CMST, Merge1 and Merge2  were compared in two aspects: spatial coverage and representativeness in high latitudes.

**3.2.1 Global Coverage**

The HadSST3 has not been interpolated, while the ERSSTv5 was interpolated by EOTs (Huang et al., 2017).  We do not distinguish  the interpolated or non-interpolated boxes  and compared these boxes with HadSST3 directly in the following sections, because the interpolated ERSSTv5 data are meaningful and the final dataset contains all the interpolated values.

 In Figure 4, we  found that the spatial coverage of Merge2 increases steadily with time

from January 1900 to December 2017. In contrast, in the early and middle of 20th century, the

coverage of Merge1 changed dramatically with time, and became steady and close to Merge2 after the

late of 20th century.  Noted that if the ERSSTv5 original data  was

used (Merge2_obs), the coverage would be comparable with that of Merge1  for the whole period. In

5    addition, Table3 also shows the global coverages of Merge1 and Merge2. The maximum coverage

was  found in February 1988 for Merge1 and in January 2000 for Merge2. The minimum

coverage  was found in Apr 1900 for Merge 1 and  June 1900 for Merge2. The mean coverage

[revised manuscript text omitted]
  GISTEMP1 using the baseline period from 1951 to 1980 while CMST was using the period of 1961 to 1990. Therefore, GISTEMP1 reserved more

term stations in 1951-1980.

 Figure 7 shown that HadCRUT4 and NOAAGlobalTemp have two minimum coverage in around 1918 and 1943/1944. However, CMST and GISTEMP1 do not have these minima. Similar to the Section 3.3, we calculated the data coverage in five latitude zones and noticed that the data coverages of HadCRUT4 and NOAAGlobalTemp have the greater fluctuations in the latitude zones of the 30 °N — 30 °S and 30 °S — 60 °S.  In order to find the latitude zone with the greatest impact on global coverage in 30 °N — 60 °S, we divided these latitude zones into 20 °N — 10 °S, 10 °N — 20 °S, 0 °— 30 °S, 10 °S — 40 °S, and 20 °S — 50 °S. It is found that the minimum value of 30 °S-60 °S coverage is the smallest, which has the greatest impact on global coverage. Therefore, the reason for small spatial coverage of HadCRUT4 and NOAAGlobalTemp is mainly due to the small coverage of the latitude zone of 30 °S — 60 °S.

Since the 30 °S — 60 °S latitude zone is dominated by oceans, the change of ST coverage in the 30 °S — 6 0 °S latitude zone is likely related to the change of SST coverage. This result is consistent with  Vose et al. (2012)  Their study  mentioned that from the early of twentieth century to the present day, the coverage of SST increased from 30 % to 70 % and the coverage of marine data decreased significantly during the two World Wars. The decrease in the coverage of HadCRUT4 and NOAAGlobalTemp is very clear during  two World Wars period. For CMST and GISTEMP, coverage is less affected during the two World Wars period because ERSSTv5

has been interpolated in many observation missing grid boxes.

**4.2 Surface temperature trends**

The study of Li et al. (2019) showeds that the recent global mean ST warming trend since 1998 derived from CMST was slightly increases slightly comparing with the existing datasets, anddatasets and is statistically significant. And In addition, it becomes closer among the newly developed global observational data (CMST), remote sensed/Buoy network infilled dataset, and adjusted reanalysis data (Cowtan and Way, 2014; Huang J et al., 2017; Simonds et al., 2017). Similar to Li et al. (2019), the temperature trends for the period of 1900-2017 in different latitudinal belts are were compared among these datasets: GISTEMP v3 250 km-Smoothing (defined as GISTEMP1)—, GISTEMP v3 1200km-Smoothing (defined as GISTEMP2), BEST with air temperatures over sea ice (defined as BEST1), BEST with water temperatures below sea ice (defined as BEST2), NOAAGlobalTemp, HadCRUT4, and CMST (Table 4).

Firstly, the ST trends in every region were compared. The temperature trend in the Northern Hemisphere high latitude is the largest ($\geq$ 0.116 ℃/decade), and) and becomes lower in the mid-latitudes of the Northern Hemisphere, the mid-latitudes of the Southern Hemisphere, and the low latitudes. The lowest temperature trend— foundis in the high latitudes of the Southern Hemisphere.

Secondly, the differences in the STs long-term trends of STs in different latitude zones are were compared. The temperature trends with largest difference occuroccurred in the high latitudes. At the

high latitudes of the Southern Hemisphere, temperature trend is the highest in HadCRUT4 (0.114 ± 0.019 ℃/decade) and  lowest in NOAAGlobalTemp v4 (0.031 ± 0.011 ℃/decade). The largest difference between the highest and the lowest temperature trends is 0.083 ℃/decade. In the high latitudes of the Northern Hemisphere, the highest temperature trend was found in

5  GISTEMP2 (0.164 ± 0.014 ℃/decade) and  lowest in CMST (0.116 ± 0.012 ℃/decade). The maximum difference is 0.048 ℃/decade. In between the middle and low latitudes, the biggest difference was found in the low latitude (0.018 ℃/decade).

Finally, the uncertainty range of the temperature trend of each dataset in different latitudes were compared. The uncertainty of every dataset is very small in the middle and low latitudes, and the largest

10  uncertainty was in the high latitudes. In the high latitudes of the Southern Hemisphere, the uncertainty in CMST is the smallest. In the Northern Hemisphere high latitudes, the uncertainty in CMST is larger than  BEST2 but smaller than other datasets.

**4.3 Inter-annual variations**

Figure 9 shows the area-weighted averaged time series of the global ST anomalies for the period

15  1990-2017 in seven datasets The temperature anomalies showed a clear warming trend from 1900 to 2017. For CMST, the highest temperature anomaly is 0.82 ℃ in 2016. There is a significant warming trend from  1910s to  1940s and  1960s to 2017. In contrast, there is a cooling trend  during

1940s to 1950s. The changes were highly consistent with the other datasets and are related to the changes of El Niño and La Niña events, volcanic eruptions, sea ice cover, and other factors (Simmons et al., 2017). Overall, the global ST changes in CMST and other datasets are similar over the period of 1900-2017. From 1920s to 1970s, CMST is slightly lower whereas HadCRUT4 is slightly higher than other datasets. The maximum difference in between CMST and HadCRUT4 is in 1938 and 1948, and the difference in temperature anomalies within these two years is 0.18 ℃. In 1938, the temperature anomalies are -0.17 ℃ and 0.01 ℃ in CMST and HadCRUT4, respectively. In 1948, the temperature anomalies are -0.20 ℃ and -0.02 ℃ in CMST and CRUT4, respectively.

The time series of ST anomalies in the seven datasets are also divided into the Northern Hemisphere (a), the Southern Hemisphere (b), and five latitudinal zones in 90 °N – 60 °N (c), 60 °N – 30 °N (e), 30 °N – 30 °S (g), 30 °S – 60 °S (f), and 60 °S – 90 °S (h). Results clearly showed the time series of temperature anomalies in every dataset is highly consistent in the Northern Hemisphere (Figure 10a). At the low latitudes (Figure. 10e, f, g), the maximum ST of several datasets occurs in 2016, whereas the minimum occurs in different years. The minimum ST was found in 1917 for most of the datasets (GISTEMP1, GISTEMP2, BEST1, BEST2, HadCRUT4, and CMST), but it was found in 1908, 1909, and 1910 in NOAAGlobalTemp (Figure 10a).

In the mid-latitude zone (Figure 10e, f), the times with maximum ST in seven datasets are

generally consistent. The maximum ST occurs in 2015 in the 60 °N — 30 °N and in 2017 in the 30 °S — -60 °S. The times with the minimum ST appear to be the same in seven datasets. The minimum ST  was detected in 1912 in the 60 °N — -30 °N, and in 1911 in the 30 °S — -60 °S. In the high latitudes of the Northern Hemisphere, the maximum ST consistently occurs in 2016, and the minimum

5 ST consistently  happened in 1902.

In the high latitudes of the Southern Hemisphere (Figure. 10d), the CMST is consistent with all the series derived from other datasets after 1960. There are many less stations/grid boxes in the Antarctic/higher latitudes, hence  larger variances were found before 1960.

**5. Data Availability**

10 The datasets used in CMST were derived from published data by NHMS (China, Russia, USA, Canada, Australia, some Asian countries, etc.) or climate data research institutions (UK/CRU, NOAA/NCEI). Part of the data are exchanged from some countries or regions, and therefore is conditionally available to public. Details of the data sources are as follows: The C-LSAT 1.3 in gridded form with a resolution of 5 ° ×× 5 ° developed by SUN Yat-Sen University (SYSU) & China

15 Meteorological Administration is available on the Climate Explorer website of the Royal Netherlands Meteorological Institute (KNMI) (http://climexp.knmi.nl/select.cgi?id=someone@somewhere&field=clsat_tavg). ERSST.v5 is from the

US                                   NOAA/NCEI                                   at

https://www.ncdc.noaa.gov/data-access/marineocean-data/extended-reconstructed-sea-surface-temperat ure-ersst-v5. The China Merged Surface Temperature Data (CMST) dataset developed by SYSU is currently public released on the Climate Explorer website of the Royal Netherlands Meteorological Institute (KNMI) (http://climexp.knmi.nl/select.cgi?id=someone@somewhere&field=cmst ). With the digital object identifiers (DOIs) (https://doi.pangaea.de/10.1594/PANGAEA.901295) issued for the data sets (Yun et al., 2019), we hope to have provided a repository of a new global ST analyses across the past 120 years from present backnow to year 1900, for the public as well as for the scientific user community as well as the public.

**6. Conclusion**

A new global ST dataset of CMST (China Merged Surface Temperature) has been developed based on the LSAT dataset (C-LSAT1.3) and SST dataset (ERSSTv5). Based on the LSAT dataset (C-LSAT1.3) and SST dataset (ERSSTv5), a new global ST dataset of CMST (China Merged Surface Temperature) has been developed. This dataset was completed by the cooperation between Sun Yat-sen University (SYSU), China Meteorological Administration (CMA), and the United States NOAA/NCEI. In CMST, we found:

1) The spatial coverage is become larger when C-LSAT1.3 and ERSSTv5 are merged., and It is

smaller when C-LSAT1.3 and HadSST3  are merged,  particularly in the Polar Regions.  Besides that, 
[revised manuscript text omitted]

---

## Referee Comment (RC2) · Anonymous Referee #2 · 10 Aug 2019

Review ESSD-2019-80, CMST

The authors merged C-LAST1.3 (land surface temperatures) with two well-known ocean SST products, ERSSTv5 and HadSSTv3, They documented blending techniques used to combine land and ocean data. After comparisons, they indicate C-LAST1.3 with ERSSTv5 as the preferred combination to produce China Merged Surface Temperature (CMST) data product. They show that CMST looks very reliable compared to other global surface temperature data products. The CMST data seem easily accessible via Pangaea and KNMI.

This reviewer regards CMST as a useful new product. Publication in ESSD seems appropriate. However, manuscript, what ESSD calls 'data description' needs substantial changes to shorten and refine.

Authors should use ESSD to describe and promote CMST. Nothing more. No histories of SST products, no evaluation of others' data products (e.g. BEST), just what they did to produce and validate CMST.

The authors present many users with a contradiction: description and production of an openly accessible global product, CMST, based primarily on a land product C-LAST1.3 whose description (Xu et al. 2018) remains behind a paywall. Everything user might want to know about homogenization, outlier detection and general quality control of the land product, e.g. as CRUTEM4 provided in ESSD 2014, https://doi.org/10.5194/essd-6-61-2014, remains in this case in separate non-open Climate Dynamics paper. That CMST represents a quality outcome, e.g. as in Figures 9 and 10 of this paper, seems obvious. The quality control procedures authors used to produce C-LAST remain out of reach for many users.

Unless CMST data explicitly resolve or refute hiatus or other global data oddities, user does not need to read those descriptions here. Focus on CMST, how authors built it, and its validation.

Likewise, authors do not need repeat evaluation procedures for other components they did not produce. User does not need to read here how other land ST or SST data sets evolved. We can read those histories and comparisons elsewhere. Again, unless some text pertains uniquely to CMST, user does need to see that text here.

Authors should completely avoid BE problems. They have raised valid issues, responded with useful and appropriate detail in the discussion. The issue now appears to lie with BE. Unless authors have separate information from BE researchers (in which case BE or these authors should make that information open via the discussion thread), these authors should remove BE products. Not the task for these authors to fix BE problems. Remove last column of Table 1. Remove Figure 2. When these authors reference the BE product, e.g. on line 17 of page 8, they should simply insert a footnote, something like 'despite careful efforts we could not extract reliable information from BEST. Authors focus on CMST quality control, not on BE problems.

Manuscript contains several additional redundancies. Authors should follow closely the rule: if text helps users understand CMST, keep it. If text does not pertain specifically to CMST, remove it. Authors could produce a good ESSD data description focused on CMST.

---

## Author Response (AR1)

Review ESSD-2019-80, CMST

This manuscript devoted to generating a new merge of global surface temperature anomaly dataset, through merging C-LSAT and ERSSTv5 datasets. However, there are several problems in the manuscript.

1. The global monthly temperature anomaly product of 5 degree seems to be too coarse to serve for investigation of temperature variation, especially for some regions such as the land and ocean convergence zones. Currently, development of climatic dataset should provide data of higher spatial resolution.

Response:

Thank you very much for the suggestions.

This dataset - CMST, is still a global surface temperature dataset used to analyze the surface Temperature change in large scales since the start of 20 century. In this perspective, 5 degree would be enough for the large scale (global/hemispheric/continental) mean ST change due to the good representativeness of surface temperature. The similar global ST datasets like HadCRUT, NOAAGlobalTemp, GISTem are all currently in 5 degree resolution.

As you mentioned, higher resolution datasets are very important for regional/local surface temperature variation analysis. In fact, we are developing the higher resolution global land surface air temperature dataset with advanced interpolation methods. But it is also very difficult to develop a high resolution dataset in good quality. Many factors like location, elevation, coastal proximity, topographic facet orientation, vertical atmospheric layer, topographic position, and orographic effectiveness of the terrain should be considered, especially in the construction of the climatology. (Daly et al, 2008). There are some high resolution global land surface air temperature datasets: CRU T4.03 (0.5*0.5), Berkerly Earth (1*1) (Rohde et al., 2013), and WorldClim (only climatology). CRU T4.03 (New et al., 2002) and WorldClim (Fick and Hijmans, 2017) is based on AUSPLINE (climatology) +IDW (anormaly), BE is based on Kriging. But we need to point out that the high resolution data need to pay more attention to the spatial distribution (lower RMSE, AME) rather than the temporal changes, which may lead some inhomogeneities in the grid series and thus affect the detection of the long-term trends. So this would be a different concern in some degree, we will discuss this in the future studies.

Ref:

Daly, C., Halbleib, M., Smith, J.I., Gibson, W.P., Doggett, M.K., Taylor, G.H., Curtis, J., and Pasteris, P.A. 2008. Physiographically-sensitive mapping of temperature and precipitation across the conterminous United States. International Journal of Climatology, 28: 2031-2064.

New M, Lister D, Hulme M, Makin I. 2002. A high-resolution data set of surface climate over

   global land areas. Climate Res. 21: 1–25.

Fick S. E., R. J. Hijmans, 2017, WorldClim 2: new 1-km spatial resolution climate surfaces
   for global land areas. International Journal of Climatology, 37: 4302–4315.
   https://doi.org/10.1002/joc.5086

Rohde R., R. A. Muller, et al. (2013) *Berkeley Earth Temperature Averaging* Process.
   Geoinfor Geostat: An Overview 1:2. doi:10.4172/gigs.1000103

2. In Page 12 and Figure 3, to calculate surface temperature anomaly at 5-degree scale, the authors adopted averaged value of temperature anomaly at 1-degree scale. Moreover, spatial transformation of sea surface temperature anomaly from 2 to 1 degree is not robust. Some geographic statistic methods should be employed to guide these spatial transformations of geographic data, such as Nearest Neighbor, Bilinear, Kriging, and Inverse Distance Weighting interpolations.

Response:

   Thank you very much for the suggestions.

   You are right when one does the similar downscaling in climatology (climatic normal) field. Due to the spatial inhomogeneity of sea surface temperature (SST), there will be a certain difference when different methods are used. But it will be a little different when downscaling the temperature anomalies (SSTA/SATA). The anomalies always have much better spatial representation especially in SSTA. Here in this manuscript, the purpose of downscaling the SSTA to 1*1 resolution is only for the convenience of merging the land and marine data. We downscaled the 2*2 sea surface temperature anomaly to the 1*1 resolution to match with the land surface air temperature data (C-LSAT) in 1*1 resolution (downscaled in the same method from 5 degree grid box). The slight difference between the simple averaged values with the interpolated values with geographic statistic methods will not matter much in the final CMST dataset. Based on above, we only simple adopt the average values for the SST anomalies downscaling.

3. I noticed that the dataset was stored as txt files. It is better to give product in NetCDF format to provide the longitude and latitude information. In addition, in the "readme.txt", authors stated that -999.99 was set as the missing value; however, I noticed that 999.99 and NaN were used as the missing value in the specific data files.

Response:

   Thank you very much for the suggestions.

   We have transferred the ASCII code data into the NetCDF and updated the readme.txt. We will update them on the website whenever we can.

4. It is necessary to reorganize the statement of each part and find a professional English Editor to improve the language quality of the ms, because the whole ms is rough to be read.

Response:

   Thank you very much for the suggestions.

   We have revised the ms as much as we can. And the revised manuscript has been read through for more than one time by the coauthors and some friends from English speaking areas, and we think the revised manuscript is much better.

**Anonymous Referee #2**

Authors could produce a good ESSD data description focused on CMST. Suggestions in supplement.

Please also note the supplement to this comment:https://www.earth-syst-sci-data-discuss.net/essd-2019-80/essd-2019-80-RC2-supplement.pdf

Review ESSD-2019-80, CMST

The authors merged C-LAST1.3 (land surface temperatures) with two well-known ocean SST products, ERSSTv5 and HadSSTv3, They documented blending techniques used to combine land and ocean data. After comparisons, they indicate C-LAST1.3 with ERSSTv5 as the preferred combination to produce China Merged Surface Temperature (CMST) data product. They show that CMST looks very reliable compared to other global surface temperature data products. The CMST data seem easily accessible via Pangaea and KNMI.

This reviewer regards CMST as a useful new product. Publication in ESSD seems appropriate. However, manuscript, what ESSD calls 'data description' needs substantial changes to shorten and refine. Authors should use ESSD to describe and promote CMST. Nothing more. No histories of SST products, no evaluation of others' data products (e.g. BEST), just what they did to produce and validate CMST. The authors present many users with a contradiction: description and production of an openly accessible global product, CMST, based primarily on a land product C-LAST1.3 whose description (Xu et al. 2018) remains behind a paywall. Everything user might want to know about homogenization, outlier detection and general quality control of the land product, e.g. as CRUTEM4 provided in ESSD 2014, https://doi.org/10.5194/essd-6-61-2014, remains in this case in separate non-open Climate Dynamics paper. That CMST represents a quality outcome, e.g. as in Figures 9 and 10 of this paper, seems obvious. The quality control procedures authors used to produce C-LAST remain out of reach for many users.

Response:

Thanks for the positive comments and the suggestions.

What we have done for the revision of the manuscript includes:

1) We added the text about the integration, data quality control and homogenization, which mostly has been detailed described in Xu et al. (2018) in Climate Dynamics (This is a publicly issued scientific Journal, yet it is still behind a paywall at present), so we only briefly repeated the most important information to avoid the duplicate of the paper.

2) We deleted most of the texts on the histories of SST products and the evaluation of others' data products. We read the CRUTEM4 paper carefully, and revised the manuscript with the reference of the paper. Now the manuscript has been recognized and shortened according to the purpose of the description of the data set.

3) We recognized the comparisons of the CMST with other existing datasets in three aspects: special coverage and anomalies and trends. And the texts have been shortened.

Unless CMST data explicitly resolve or refute hiatus or other global data oddities, user does

not need to read those descriptions here. Focus on CMST, how authors built it, and its validation.

Response:

Thanks for the suggestion. We deleted the hiatus part.

Likewise, authors do not need repeat evaluation procedures for other components they did not produce. User does not need to read here how other land ST or SST data sets evolved. We can read those histories and comparisons elsewhere. Again, unless some text pertains uniquely to CMST, user does need to see that text here.

Response:

Thanks for the suggestion. We deleted the histories of the land ST and SST datasets.

Authors should completely avoid BE problems. They have raised valid issues, responded with useful and appropriate detail in the discussion. The issue now appears to lie with BE. Unless authors have separate information from BE researchers (in which case BE or these authors should make that information open via the discussion thread), these authors should remove BE products. Not the task for these authors to fix BE problems. Remove last column of Table 1.Remove Figure 2. When these authors reference the BE product, e.g. on line 17 of page 8, they should simply insert a footnote, something like 'despite careful efforts we could not extract reliable information from BEST. Authors focus on CMST quality control, not on BE problems.

Response:

Thanks for the suggestion. We removed the BE products from table 1 and figure 2, and added the footnote saying "Despite careful efforts we could not extract reliable information from BEST. So the comparison with BEST has been deleted according to the reviewer's suggestion".

Manuscript contains several additional redundancies. Authors should follow closely the rule: if text helps users understand CMST, keep it. If text does not pertain specifically to CMST, remove it. Authors could produce a good ESSD data description focused on CMST.

Response:

Thanks for the suggestion. We made several revisions based on the reviewer's suggestions, which makes the manuscript more readable than before.

**A new merge of global surface temperature datasets since the start of the 20th Century**

Xiang Yun[1,2], Boyin Huang [3], Jiayi Cheng[1#], Weihui Xu[4], Shaobo Qiao[1#], Qingxiang Li [1,#*]

1 *School of Atmospheric Sciences and Guangdong Province Key Laboratory for Climate Change and Natural Disasters, SUN Yat-Sen University, Guangzhou, China*

2 *Chinese Academy of Meteorological Sciences, CMA, Beijing, China*

3 *National Centers of Environmental Information, NOAA, Asheville, USA*

10    4 *National Meteorological Information Center, CMA, Beijing, China*

**Southern Laboratory of Ocean Science and Engineering (Guangdong Zhuhai), Zhuhai, China**

*Corresponding to:* Qingxiang Li (liqingx5@mail.sysu.edu.cn)

**Abstract**

Global surface temperature (ST) datasets are the foundation for global climate change research.

15    Several global ST datasets have been developed by different groups in NOAA/NCEI,NASA/GISS,  UKMO Hadley Centre & UEA/CRU and Berkeley Earth. In this study, a new global ST dataset was presented named China Merged Surface Temperature (CMST) dataset. CMST is created by merging the China-Land Surface Air Temperature (C-LSAT1.3) with sea surface temperature (SST) data from the Extended Reconstructed Sea Surface Temperature version 5 (ERSSTv5). The merge of C-LSAT and

20    ERSSTv5 shows a high spatial coverage extended to the high latitudes and is more consistent with a reference of multi-datasets average in the Polar Regions. Comparisons indicated that CMST is

consistent with other existing global ST datasets in interannual-decadal variations and long-term trends at global, hemispheric, and regional scales from 1900 to 2017. The CMST dataset can be used for global climate change assessment, monitoring, and detection. The CMST dataset presented here is publicly available at: https://doi.pangaea.de/10.1594/PANGAEA.901295 (Yun et al., 2019) and has been

5 published on the Climate Explorer website of the Royal Netherlands Meteorological Institute (KNMI) at:

http://climexp.knmi.nl/select.cgi?id=someone@somewhere&field=cmst.

**1. Introduction**

The long-term trend of the global mean surface temperatures (GMST) is one of a common measure in observing the change of climate. Therefore, the biases of the observed surface temperature (ST) dataset, particularly the sampling bias of high latitudes stations, has received much attention in the past few years (Cowtan and Way, 2014; Jones, 2016; Li et al., 2017; 2019; Simonds et al., 2017; Huang J. et al., 2017a). The optimization and improvement of observed climate data is a long-term task, as a reference base for climate change research and verification benchmark for other climatic data products.

A total of four global land surface air temperature (LSAT) observation series and three global ST series were exhibited by The Intergovernmental Panel on Climate Change (IPCC, 2013) few years ago. These four LSATs includeing the Climatic Research Unit (CRU) land surface air temperature, version 4 (CRUTEM4; Jones et al., 2012), Global Historical Climatology Network-monthly (GHCNm) temperature, version 3 (GHCNm v3; Lawrimore et al., 2011), Goddard Institute for Space Studies analysis of land surface air temperature (GISS; Hansen et al., 2010), and Berkeley Earth Surface Temperature group land temperature (Berkeley; Rohde et al., 2013). While the three global ST series are the Met Office Hadley Centre and Climatic Research Unit Temperature version 4 (HadCRUT4; Morice et al., 2012), Merged Land–Ocean Surface Temperature (MLOST; Vose et al., 2012b), and Goddard Institute for Space Studies Surface Temperature Analysis (GISTEMP; Hansen et al., 2010).

All these datasets indicated that the Earth has experienced a "warming hiatus" period from 1998 to 2012.and this issue has been attracted many attention from the researchers around the world. However, by analyzing the sea surface temperature (SST) and global ST from the National Oceanic and Atmospheric Administration / National Centers for Environmental Information (NOAA/NCEI), Karl et al. (2015) suggested that the "warming hiatus" is due to the artifact of the data processing. Besides that, Lewandowsky et al. (2015) noted that this short-term warming trend "hiatus" is a conditional statistical artifact and it is not a real scientific fact. After correcting the sampling biases of the temperature data over the Arctic region from few studies by using reanalysis data (Simonds et al., 2017), satellite remote sensing data (Cowtan and Way, 2014), and Arctic buoy data (Wang et al., 2017), their results have come to a similar conclusion.

These global ST data products have been updated over the past few years since the publication of IPCC (2013). For instance, the NOAA has updated the Extended Reconstructed Sea Surface Temperature (ERSST) version 3 to ERSSTv4 (Huang et al., 2015) and ERSSTv5 (Huang B. et al., 2017), updated LSAT dataset GHCNm v3 to GHCNm v4 (Menne et al., 2018), and renamed MLOST to NOAA Global Surface Temperature (NOAAGlobalTemp). The GISTEMP has been updated its SST component to ERSSTv5 (Huang B. et al., 2017b). While CRUTEM has been updated to CRUTEM4.6. The Met Office updated the Hadley Centre SST to version 3 (HadSST3) using the median of 100 ensemble members. And lastly, the Berkeley team used the median of the HadSST3 ensemble

ensemble to form the Berkeley Earth Surface Temperature (BEST) dataset.

The products' updates are based on the advanced knowledge of data analysis methodology or improved data availability. In general, the GMST has continuously been improved by the increased number and area coverage of observational data over the land (LSAT) and oceans (SST). There are two aspects to improve the LSAT datasets. Firstly, increase the data coverage  and density of stations  especially in the key areas with sparse observations. For example, the number of observations is increased in both C-LSAT (Xu et al., 2018) and GHCNm v4 (Menne et al., 2018) using a newly released International Surface Temperature Initiative (ISTI) datasets (Thorne et al., 2011) or datasets through regional cooperation with Asian countries such as Vietnam and South Korea. Coverage of datasets increases with larger number of observations and hence reducing the sampling biases, particularly for high latitudes area (Polar region) and observation-sparse regions (such as South America and Africa). Next, the second aspect is improving the accuracy of regional climate changes. For example, the latest C-LSAT (Xu et al., 2018) has integrated more regional homogenization results, especially over the China (Li et al., 2009; Xu et al., 2013), East Asia, Europe, Australia (Trevin, 2013), and Canada (Vincent et al., 2012).

On the other  hand, there are also two aspects to improve the SST datasets: (1) integration of much better raw observational data and; (2) replacing a single analysis to multi-member ensemble analyses. For instances, ERSSTv5 is using the most recently available International Comprehensive

Ocean-Atmosphere Data Set release 3.0 (ICOADS R3.0; Freeman et al., 2017), optimized climate models and more accurate buoy data in adjusting the ship data. Meanwhile, HadSST3 introduces a variety of bias correction models and median SST of the 100 ensemble members was used as the best estimation.

5 ~~Among all the existing global ST datasets (e.g.,,, HadCRUT and NOAAGlobalTemp), the merging methods in combining the land and ocean datasets basically is very similar to each other. The merging process of HadCRUT includes: First, the land and ocean data were processed into 100 ensemble datasets according to the bias evaluation parameters, in which the anomaly values calculated separately for each grid box. Then, anomalies in the grid boxes of the land and ocean boundary weighted by the~~

10 ~~fraction of land and ocean areas. If the land area covers less than 25 %, it is calculated as 25 %. If there is a measured SAT anomaly in a grid box covered with sea-ice, The SAT will be used to represent the SST anomaly. The HadCRUT ensemble dataset has a reference period range from 1961 to 1990 with a resolution of 5 ° × 5 ° (Morice et al., 2012). There are three steps of merging process in NOAAGlobalTemp. The first step is to identify theLSAT/SST low frequency changes by~~

15 by the

This study presents a new merged global ST dataset based on the recently developed C-LSAT and the latest ERSSTv5 using a method which is similar to the HadCRUT and NOAAGlobalTemp, providing a new reference to the climate or climate change studies. The remainder of this paper is arranged into different sections as below. The land and ocean datasets and their updates are briefly introduced in section 2. The merging process of CMST is given in section 3. Section 4 discussed the comparisons of CMST with other existing ST datasets. The availability of the resulting dataset (Yun et al., 2019) is reported in section 5 and summary of results are presented in section 6.

**2. Updates of land and ocean data**

**2.1 Land surface air temperature data**

**2.1.1 Data sources of C-LSAT**

The C-LSAT1.0 dataset (Xu et al., 2018) processed the SAT data since 1900 from a total of 14 data sources, these including three global data sources (CRUTEM 4.6, GHCNv3, and BEST), three regional data sources from Scientific Committee on Antarctic Research (SCAR), daily dataset for European Climate Assessment (ECA&D), and Historical Instrumental climatological Surface Time series of the greater Alpine region (HISTALP), and eight national data sources from China, America, Russia, Canada, Australia, Korea, Japan, and Vietnam.

The C-LSAT version 1.3 dataset is used in this study. Compared to the C-LSAT version 1.0  , the version 1.3 is updated to December 2017. According to Xu et al. (2018), national, regional and global datasets are ranked as higher, middle and lower priorities, respectively. Based on the priority of the data resources, a total of 4917 high priority stations were added, while total of 1364 low priority stations were deleted (replaced by higher priority stations). Most of the newly added raw data (they are not the real "raw" data, most of them have been quality controlled) were obtained from the International Surface Temperature Initiative (ISTI) Projects and have been homogenized through the same approach as Xu et al. (2018). The distribution of these extra 3553 stations shown in Figure 1. According to Xu et al. (2018), the C-LSAT version 1.0 had some advantages over the existing global LSAT datasets in station number and spatial coverage. Thus, the current C-LSAT version 1.3 has more station number than the existing datasets in many regions over the global land surface. Figure 1 shows the extra stations compared to version 1.0 and Table 1 shows the comparison of the station number for different datasets, indicating an enhanced coverage and distribution/sampling of LSAT observations.

**2.1.2 Integration, QC and homogenization of C-LSAT**

All of the data sources collected were firstly merged into a single comprehensive dataset. The merge process was based on metadata matching and data equivalence criteria. Each candidate station was compared to all highest priority (target) stations in two steps. In the first step, each candidate

station was run through all the target stations and two metadata criteria were calculated for identifying matching stations: when 1) the distance of two separate stations (same name) fall within 5km and the height difference fall within 50m, and 2) the Jaccard Index (JI) (Jaccard 1901; Xu et al. 2018) for two stations (different names) reaches 0.8, they  passes the match test; In the second step, a data comparison was made between the same stations from different sources using the index of agreement (*IA*) (Willmott 1985; Xu et al., 2018). If the *IA* reaches 0.8, the candidate station is merged with the target station. However, the lower-priority source is used  where the higher-priority source is unavailable.

Similar to GHCN-V3 (Lawrimore et al., 2011), a three-step quality control (QC) process has been used for the merged dataset. QC 1: Check for climate anomalies. Anomalies higher than five times the standard deviation of the monthly mean at each station are treated as missing; QC 2: Check for spatial consistency. At a given time, $\left| Z_i - \overline{Z_{ij}} \right| > 3.5\sigma_{ij}$ or $\left| Z_i - \overline{Z_{ij}} \right| > 2.5$ is considered as an outlier and excluded (where $Z_i$ is the normalized (to the baseline period of 1961-1990) air temperature at the target station, $Z_{ij}$ is the normalized air temperature at the neighboring stations (not exceeding 20) within 500 km from the target station, $\overline{Z_{ij}}$ is the mean of normalized air temperature at the neighboring station, and $\sigma_{ij}$ is the standard deviation of normalized air temperature at the neighboring station); and QC 3 is the check for internal consistency (to ensure $T_{max} > T_m > T_{min}$ for the same month, since the internal inconsistency may arise when the station data has been integrated from different

sources). The QC results shows problematic data from 54 (QC1), 349 (QC2) and 1544 (QC3) station months has been detected out and treated as missing values. Although the proportion is relatively low (totally less than 0.02%), the impact of QC process on dataset products is significant.

At last, Two steps were taken to ensure the homogeneity of the station time series: (1) tThe data series from the existing national homogenized datasets were directly integrated into C-LSAT without any change, which is approximately 50 % of the stations in C-LSAT. The benefit of including existing data sets doing this way is to improve the accuracy of regional climate change estimates by using several regional homogenized datasets developed by the corresponding NMHS (National Meteorological and Hydrological Services) or Climatic Data Centers. We believe that detailed metadata and the expert knowledge will be helpful to generate better regional homogenized datasets, and the differences induced by using different homogenization methods are less important comparing with the differences in observing practices, instrumentation, and post-observation processing used by different nations/countries (Xu et al., 2018;). (2) Tthe inhomogeneities in the rest of the station series ($T_m$ from about 6500 stations) were detected and adjusted with the penalized maximal t-test method (Wang et al., 2007; Xu et al., 2018). The benefit of doing this way is to improve the accuracy of regional climate change estimates by using several regional homogenized datasets developed by the corresponding NMHS (National Meteorological and Hydrological Services) or Climatic Data Centers. We believe that detailed metadata and the expert knowledge will be helpful to generate better regional homogenized

   The process is totally the same with what they have done in Xu et al. (2018). ~~The C-LSAT version 1.3 dataset is used in this study. Compared to the C-LSAT version 1.0 range from 1900 to 2014 (Xu et al., 2018), the version 1.3 is updated to December 2017. According to Xu et al. (2018), national, regional and global datasets are ranked as higher, middle and lower priorities, respectively. Based on the priority of the data resources, a total of 4917 high priority stations were added, while total of 1364 low priority stations were deleted. Most of the newly added raw data were obtained from the International Surface Temperature Initiative (ISTI) Projects and have been homogenized through the same approach as Xu et al. (2018). The distribution of these extra 3553 stations shown in Figure 1. According to Xu et al. (2018), the C-LSAT version 1.0 had some advantages over the existing global LSAT datasets in station number and spatial coverage. Thus, the current C-LSAT version 1.3 has more station number than the existing datasets in many regions over the global land surface. Figure 1 shows the extra stations compared to version 1.0 and Table 1 shows the comparison of the station number for different datasets, indicating an enhanced coverage and distribution/sampling of LSAT observations.~~

For the comparison purposes, other LSAT datasets were collected from CRUTEM4 (https://www.metoffice.gov.uk/hadobs/), GHCNm v3 (ftp://ftp.ncdc.noaa.gov/pub/) and Berkeley Earth (BE) (http://www.berkeleyearth.dev/)[1]. All these datasets above were downloaded in August 2018. The following calculations are based on stations with a time length greater than 15 years in between 1900 to 2017.

From a global and hemispheric perspective, the C-LSAT version 1.3 dataset has more stations than the other datasets in the Global and Southern Hemisphere (SH) (Table 1). Besides that, C-LSAT also have the largest stations number for seven regions - Asia, Africa, Australia, South America, Europe, Antarctic, and Arctic as defined in Xu et al. (2018). ~~The only exception was in North America, where BE has the most stations number. However, for BE dataset, stations from North American account for 85.7 % of those from the Northern Hemisphere, this meaning that the stations from other parts of the Northern Hemisphere is only 14.3 %. While for C-LSAT dataset, stations from North America account for 30.7 % of the Northern Hemisphere, and those from other parts of the Northern Hemisphere account for 69.3 % (Figures 2a and 2b). Furthermore, when the number of effective grid boxes in $5° \times 5°$ grid containing observations are calculated between 1900 and 2017, we noticed that C-LSAT has more effective grid boxes even though the Berkeley dataset has more stations number. In other words, although the Berkeley's stations number in the Northern Hemisphere is slightly higher than C-LSAT, the~~
* * *
[1] Despite careful efforts we could not extract reliable information from BEST. So the comparison with BEST has been deleted according to the reviewer's suggestion.

**2.2 Sea surface temperature data**

In general, only in-situ observational data were used when merging LSAT and SST for the

commonly-used global ST datasets. For example, HadCRUT4 and BE used HadSST3 (the median of

100 ensemble datasets), meanwhile the NOAAGlobalTemp and GISTEMP were using

ERSSTv5. Both HadSST3 and ERSSTv5 datasets use only in-situ data. Other datasets, such as COBE

(Hirahara et al., 2014) and HadISST (Rayner et al., 2003) which is using both in-situ and satellite data,

were not used as a source in the merging of global ST data, although they are frequently used in SST studies. Therefore, the latter two SST datasets  were used only for comparisons in this study.

**3. Reconstruction of Global ST Dataset**

**3.1 Merging Schemes**

Generally in previous studies, the global ST dataset was merged using LSAT and  SST dataset. Among all the existing global ST datasets (e.g., HadCRUT and NOAAGlobalTemp), the merging methods in combining the land and ocean datasets basically is very similar to each other. (Morice et al., 2012;Vose et al., 2012).  In this study, C-LSAT1.3 is merged with HadSST3 and ERSSTv5 separately. The final merged global ST dataset will be selected based on the comparison of the quality of different merging schemes. These two SST datasets are reprocessed before the merging. The median of the 100-member ensemble datasets in HadSST3 were calculated for each grid box (Kennedy et al., 2011). The ERSSTv5 has a value of -1.8 ℃ in many grid boxes in the Arctic and Southern Ocean, which refers to the area where the sea ice coverage is above 90 %. Therefore, some special treatment is needed for these grid boxes. If the anomalies are 0 ℃ and SSTs are -1.8 ℃, then the

value of -1.8 ℃ in ERSSTv5 will be replace with missing values. The reference period for both HadSST3 and ERSSTv5 were taken as 1961-1990.

The two merging schemes are described as follows:

(1) Merge1: C-LSAT1.3+HadSST3 (ensemble). Giving the resolution of both two datasets are $5° \times 5°$, these two datasets were directly merged using the ratios of ocean and land surface areas in a specific grid box.

(2) Merge2: C-LSAT1.3+ERSSTv5. Since the resolution of these two datasets are different, they were unified onto the same resolution ($1° \times 1°$ resolution), and then merged using the ratios of ocean and land areas.

The merging process of C-LSAT1.3 and ERSST are described as follows (and in Figure 2):

(1) The anomalies were calculated in each grid boxes with respect to the base period 1961-1990 for C-LSAT and ERSSTv5, respectively.

(2) For the ocean-land boundary part, the fraction of land and ocean areas is considered (see Figure 3̶2, taking the January 2017 as an example). The detailed procedures are:

(a) Downscale the land (C-LSAT1.3) and ocean data to $1° \times 1°$ resolution. The resolution of the ocean data is $2° \times 2°$, which is distributed in 4 grids of $1° \times 1°$. The resolution of the land data is $5° \times 5°$, which is distributed in 25 grids of $1° \times 1°$.

(b) Using the ocean-land mask file to differentiate all grids globally into land or ocean (download

link: http://www.ncl.ucar.edu/Applications/Data/cdf/landsea.nc). The ocean-land mask file is based on

Rand's global elevation and depth data, and the resolution of the ocean-land mask is re-gridded to $1° \times 1°$. The ocean-land mask file contains five types of markers: 0 for ocean, 1 for land, 2 for lakes, 3 for islands, and 4 for ice sheets. Marine data was used in parts of the ocean and ice sheets, and land data was used in parts of land, lakes, and small islands.

    (c) The $1° \times 1°$ ocean grid data and the $1° \times 1°$ land grid data were spliced by the ocean-land mask to obtain $1° \times 1°$ global ST grid data.

    (d) The averaged surface temperature anomaly (STA) in each $5° \times 5°$ grid was calculated as:

$$STA_{(5°\times5°)}(i,j) = \frac{1}{25}\left(\sum STA_{(1°\times1°)}(ii-2:ii+2, jj-2:jj+2)\right)$$

**3.2 Comparison of two merged schemes**

    Based on the methods above, C-LSAT1.3 grid data is merging with HadSST3 and ERSSTv5 data to form the C-LSAT+HadSST (Merge1) and C-LSAT+ERSST (Merge2) global ST datasets, respectively. In order to choose a better merging scheme in CMST, Merge1 and Merge2 were compared in two aspects: spatial coverage and representativeness in high latitudes.

**3.2.1 Global Coverage**

    The HadSST3 has not been interpolated, while the ERSSTv5 was interpolated by EOTs (Huang et al., 2017). We do not distinguish the interpolated or non-interpolated boxes and compared these boxes with HadSST3 directly in the following sections, because the interpolated ERSSTv5 data are

meaningful and the final dataset contains all the interpolated values.

In Figure 4̶3, we found that the spatial coverage of Merge2 increases steadily with time from January 1900 to December 2017. In contrast, in the early and middle of 20th century, the coverage of Merge1 changed dramatically with time, and became steady and close to Merge2 after the late of 20th century.  Noted that if the ERSSTv5 original data was used (Merge2_obs), the coverage would be comparable with that of Merge1 for the whole period. In addition, T̶a̶b̶l̶e̶3̶ Table 2 also showed the global coverage̶s̶ of Merge1 and Merge2. The maximum coverage was found in February 1988 for Merge1 and in January 2000 for Merge2. The minimum coverage was found in April 1900 for Merge 1 and June 1900 for Merge2. The mean coverage was calculated between 1900 and 2017. From the Table 3̶2, the Merge2 dataset has larger data coverage than Merge1 in all Coverage Mean, Coverage Max and Coverage Min. Although the difference between the two in Coverage Max is not very large, the difference in Coverage Means and Coverage Min between two merges is very large. This suggests that the coverage is mostly smaller in Merge1 than Merge2. Therefore, although the original data coverage of HadSST3 and ERSSTv5 is similar to each other, but with the interpolation of EOTs, the later increased its coverage greatly. Thus from the perspective of overall coverage, the dataset Merge2 is superior to Merge1 (Figure 4̶3).

Furthermore, Figure 5̶4 shows the spatial coverage of the average temperature anomalies over 20 years for Merge1 and Merge2. The six panels in Figures 5̶a̶ 4a and Figures 5̶b̶ 4b correspond to the

20-year mean temperature anomaly distribution over 1900-1919, 1920-1939, 1940-1959, 1960-1979, 1980-1999 and 2000-2017, respectively. In the early of 20th century, it is clearly seen that Merge1 lacked a large range of data in the equatorial region, the western region of the SH and the high latitude zone of the SH. In the middle of 20th century, Merge1 lacked so much data in the high latitudes of the SH. Merge1 remained lacking data at the high latitudes of the SH by the end of the 20th century. In contrast, Merge2 exhibited data in global especially after 2000s. This is due to the rapidly increase in the number of observations from Argo5obs (Argo floats between 0- and 5-m depth) in between 2000 to 2006. Since 2006, the Argo5obs has maintained close to near-global coverage. In the high latitude region, the coverage of the Merge1 dataset is also smaller than that of Merge2, which may critically impact the assessment of climate over the Arctic. This is mainly because the spatial coverage of ICOADS R3.0 used in Merge2 is slightly higher than R2.5 used in Merge1, especially in the south of $60\,^{\circ}$S and north of $60\,^{\circ}$N (Huang et al., 2017). Therefore, the coverage of the Merge1 is clearly lower than Merge2, particularly in the equatorial region and SH. Therefore, with respects to the spatial coverage of each period, Merge2 has a much better spatial coverage, especially in the early of 20th century.

**3.3.2 Representativeness in high latitudes**

To accurately compare the global and regional temperature changes between Merge1 and Merge2,

the COBE2 and HadISST1 which have satellite data integrated were introduced. First, the C-LSAT1.3 and COBE2, C-LSAT1.3 and HadISST1 datasets were merged in a similar way to form Merge3 (C-LSAT+COBE) and Merge4 (C-LSAT+HadISST) datasets. Second, the monthly temperature anomalies of Merge1-4 relatively to same baseline period (1961-1990) were calculated. The arithmetic mean of the four merged datasets was calculated for monthly temperature anomalies at each grid. As we know, each merging schemes might have uncertainties caused by different SST datasets, while the ensemble mean of all the merging datasets could have the least uncertainties. Therefore, the annual mean time series is calculated from the mean monthly temperature anomalies as a benchmark (reference series) for the two schemes.

From north to the south, the global ST is divided into five latitude zones: $90\,^{\circ}$N – $60\,^{\circ}$N, $60\,^{\circ}$N – $30\,^{\circ}$N, $30\,^{\circ}$N – $30\,^{\circ}$S, $30\,^{\circ}$S – $60\,^{\circ}$S, and $60\,^{\circ}$S – $90\,^{\circ}$S. The reference series is subtracted from Merge2 and Merge1 datasets to obtain a difference series for each region. The comparison between the two schemes indicated that the difference in mid-latitude and low latitude is small (figure omitted). The difference is large in the high latitudes (Figure 6̶5). In $90\,^{\circ}$N – $60\,^{\circ}$N, the difference between Merge2 and the reference series is steadily close to the 0 line during the period of 1900-2017, while the difference between Merge1 and the reference series is colder for the period of 1900s-1920s and warmer for 1930s-1980s a̶n̶d̶ ̶ ̶a̶l̶s̶o̶and also after 1990s. In $60\,^{\circ}$S – $90\,^{\circ}$S, the time series of Merge1 (1945) started later than Merge2 (1900), and the difference between Merge1 and the reference series (blue) is

abnormally large during 1945-1960s. While the difference between Merge2 and the reference series (red) is still very small. The large difference in Figure 6b 5b may be associated with small sampling size of the difference, or small coverage of Merge1, but the Merge2 agree well with what we have expected.

The correlation coefficients between the time series of Merge1 and the reference series and between Merge2 and the reference sequence in each latitude zone were calculated. The results showed that the correlation coefficients between Merge1 (Merge2) and the reference series are similar for the globe, the Southern HemisphereH, the Northern Hemisphere (Northern HemisphereH) and the mid-low latitudes, which exceed 0.98. Compared with the reference series in the high latitude zone, Merge2 shows much more consistence than Merge1. At 60 ˚S-90 ˚S, the correlation coefficient of Merge2 (0.90) is much larger than Merge1 (0.30). While for 90 ˚N-60 ˚N, the correlation coefficient of Merge2 (0.99) is slightly larger than Merge1 (0.97).

In summary, compared to Merge1, Merge2 dataset is superior in terms of global coverage, spatial distribution and the temporal change with the reference series. The possible reason is that the ocean data used by the ERSSTv5 dataset are the latest ICOADS R3.0 data, whereas the ocean data used by the HadSST3 dataset were obtained from ICOADS R2.5. Also, the ERSSTv5 data incorporate with more observations (such as Argo5obs). Based on the analysis above, Merge2 (C-LSAT1.3+ERSSTv5) was used as the final scheme in the later sections, which is named CMST (China Merged Surface Temperature).

**4. Comparison of CMST with other existing datasets**

**4.1 Spatial Coverage**

Spatial coverage may differ among the following products due to the difference in spatial smoothing or interpolation method applied. The HadCRUT4.6.0.0 is a non-interpolated observation dataset. NOAAGlobalTemp  is first interpolated by EOTs in both LSAT and SST, and then masked according to the actual observation availability. GISTEMP  250 km-Smoothing (defined as GISTEMP1) is interpolated with a small scan radius. CMST is interpolated by EOTs in SST but no interpolation is applied in LSAT.

First, the monthly coverage is calculated by the ratio of the areas between valid grid boxes and total grid boxes in HadCURT4, NOAAGlobalTemp, CMST, and GISTEMP1 (Figure 6). Figure  6a shows that the area coverage in CMST is larger than those in other datasets in aspects of Coverage Max, Coverage Min, and Coverage Mean, particularly the Coverage Min in CMST is much larger than those in the other datasets 6a). Second, the monthly coverage is averaged to obtain the annual average. Figure  6b shown that the coverage of CMST is larger than those of the other three datasets at any time. Furthermore, the multi-year averaged coverage between 1900 and 2017 was calculated, which is 76 %, 58 %, 71 %, and 70 %, respectively, in CMST, HadCRUT4, NOAAGlobalTemp, and GISTEMP1. In other words, the coverage in CMST is not only much larger than that in the dataset without interpolation (such as HadCRUT4), but also larger than those interpolated datasets (such as

GISTEMP1 and NOAAGlobalTemp).

The reasons why the coverage of CMST is greater than those of the other datasets are as follows: The spatial coverage of land data (CRUTEM4) in HadCRUT4 is smaller than that of C-LSAT in CMST (Xu et al., 2018), and the spatial coverage of marine data (HasSST3) in HadCRUT4 is also smaller than ERSSTv5 in CMST. The higher coverage of marine data results from two aspects: (a) The ocean data (ERSSTv5) used by CMST has additional sources of Argo data and using ICOADS R3.0 which containing more ship and buoy data. (b) The ocean data of HadCRUT4 has not been interpolated, while the ocean data used by CMST was interpolated. The spatial coverage of the land dataset (GHCNm v3) in NOAAGlobalTemp is less than C-LSAT in CMST. The spatial coverage of the marine dataset (ERSSTv4) is also less than ERSSTv5, as ERSSTv5 incorporated new ICOADS data and added a decade of Argo floats data. Additionally, GISTEMP1 has the same land dataset as NOAAGlobalTemp.  its coverage is less than CMST, and its marine dataset is the same as CMST. Therefore, the spatial coverage of CMST  is  greater than GISTEMP1 .

It should be noted that, the data coverage of GISTEMP1 increases rapidly during the 1950s (Figure 6b), which is mainly due to the rapid increase in Antarctic (60 °S – 90 °S; Figure 7b). As in CMST, the station data of GISTEMP1 in Antarctic is mostly from SCAR (Hansen et al., 2010). The differences between these two datasets are GISTEMP1 using the baseline period from 1951 to 1980 while CMST was using the period of 1961 to 1990. Therefore, GISTEMP1 reserved more short-term stations within

1951-1980.

Figure 6b also shows that HadCRUT4 and NOAAGlobalTemp have two minimum coverage in around 1918 and 1943/1944. However, CMST and GISTEMP1 do not have these minima. We calculated the data coverage in five latitude zones (Figure 7a-e) and noticed that the data coverage of HadCRUT4 and NOAAGlobalTemp have the greater fluctuations in the latitude zones of the 30 °N – 30 °S and 30 °S – 60 °S. Further analysis shows that the minimum value of 30 °S-60 °S coverage is the smallest, which has the greatest impact on the global coverage. Therefore, the  minimum values of spatial coverage of HadCRUT4 and NOAAGlobalTemp are mainly due to the minimum coverage of the  30 °S – 60 °S.

Since the 30 °S – 60 °S latitude zone is dominated by oceans, the change of ST coverage in the 30 °S – 6-0 °S latitude zone is likely related to the change of SST coverage. This result is consistent with what Vose et al. (2012) mentioned that  the coverage of marine data decreased significantly during the two World Wars.  While for CMST and GISTEMP,

coverage is less affected during the two World Wars period because ERSSTv5 has been interpolated in many observation missing grid boxes.

**4.2 Inter-annual variations and trends**

Figure 8 shows the area-weighted averaged time series of the global ST anomalies for the period 1990-2017 in seven datasets. Overall, the global ST changes in CMST and other datasets are similar over the period of 1900-2017. From 1920s to 1970s, CMST is slightly lower whereas HadCRUT4 is slightly higher than other datasets. The maximum difference  between CMST and HadCRUT4 is in 1938 and 1948, and the difference in temperature anomalies within these two years is 0.18 °C. In 1938, the temperature anomalies are -0.17 °C and 0.01 °C in CMST and HadCRUT4, respectively. In 1948, the temperature anomalies are -0.20 °C and -0.02 °C in CMST and HadCRUT4, respectively. Further,the time series of ST anomalies in the seven datasets are also divided into the NH (a), the SH (b), and five latitudinal zones in 90°N - 60°N (c), 60°N - 30°N (e), 30°N - 30°S (g), 30°S - 60°S (f), and 60°S - 90°S (h). Results clearly showed the time series of temperature anomalies in every dataset is highly consistent in the NH (Figure 9a). At the low latitudes (Figure. 9e, f, g), the maximum ST of several datasets occurs in 2016, whereas the minimum occurs in different years. The minimum ST was found in 1917 for most of the datasets (GISTEMP1, GISTEMP2, BEST1, BEST2, HadCRUT4, and CMST), but it was found in 1908, 1909, and 1910 in NOAAGlobalTemp (Figure 9a). In the mid-latitude zone (Figure 9e, f), the times with maximum ST in seven datasets are generally consistent.

The maximum ST occurs in 2015 in the 60°N – 30°N and in 2017 in the 30°S – 60°S. The times with the minimum ST appear to be the same in seven datasets. The minimum ST was detected in 1912 in the 60°N – 30°N, and in 1911 in the 30°S – 60°S. In the high latitudes of the NH, the maximum ST consistently occurs in 2016, and the minimum ST consistently happened in 1902. In the high latitudes of the SH (Figure 9d), the CMST is consistent with all the series derived from other datasets after 1960. There are many less stations/grid boxes in the Antarctic/higher latitudes, hence larger variances were found before 1960.

From figure 8, the temperature anomalies showed a clear warming trend from 1900 to 2017. For CMST, the highest temperature anomaly is 0.82 °C in 2016. There is a significant warming trend from 1910s to 1940s and 1960s to 2017. In contrast, there is a cooling trend during 1940s to 1960s. The changes were highly consistent with the other datasets and are related to the changes of El Niño and La Niña events, volcanic eruptions, sea ice cover, and other factors (Simmons et al., 2017). Li et al. (2019) showed that the ST warming trend during 1998-2012 derived from CMST was slightly increases comparing with the existing datasets and is statistically significant. It becomes closer among the newly developed global observational data (CMST), remote sensed/Buoy network infilled dataset, and adjusted reanalysis data (Cowtan and Way, 2014; Huang J et al., 2017; Simonds et al., 2017). 4.2 Surface temperature trends

The study of Li et al. (2019) showed that the recent global mean ST warming trend since 1998

derived from CMST was slightly increases comparing with the existing datasets and is statistically significant. In addition, it becomes closer among the newly developed global observational data (CMST), remote sensed/Buoy network infilled dataset, and adjusted reanalysis data (Cowtan and Way, 2014; Huang J et al., 2017; Simonds et al., 2017). Similar to Li et al. (2019), tThe temperature trends for the period of 1900-2017 in different latitudinal belts were compared among these datasets: GISTEMP v3 250 km-Smoothing (defined as GISTEMP1), GISTEMP v3 1200km-Smoothing (defined as GISTEMP2), BEST with air temperatures over sea ice (defined as BEST1), BEST with water temperatures below sea ice (defined as BEST2), NOAAGlobalTemp, HadCRUT4, and CMST (Table 43).

Firstly, the ST trends in every region were compared. The temperature ST trend in the Northern Hemisphere NH high latitude is the largest (≥0.116 ±0.012 °C/decade), and lower in the mid-latitudes of the Northern HemisphereNH (0.098 ±0.006 °C/decade), the mid-latitudes of the Southern H (0.080 ± 0.003 °C/decade)emisphere, and the low latitudes (0.082 ± 0.005 °C/decade), and The the lowest temperature trend found in the high latitudes of the Southern HemisphereSH (0.046 ±0.004 °C/decade) in CMST.

Secondly, the differences in the STs long term trends in different latitude zones were compared. The temperature ST trends with largest difference among different datasets occurred in the high latitudes (from 0.031 ± 0.011 °C/decade in NOAAGlobalTemp to 0.114 ± 0.019 °C/decade in

HadCRUT4), which shows the larger uncertainties in South Polar Region . In the high latitudes of the NH, the highest  ST trend was found in GISTEMP2 (0.164 ± 0.014 ℃/decade) and BEST1 (0. 149 ± 0. 016 ℃/decade) and lowest in CMST (0.116 ± 0.012 ℃/decade), while the former two are believed to be overestimated due to the use of air temperature over sea ice in Polar Region. The differences are lower in  the middle and low latitudes, the trends in CMST are all between the maximum and the minimum in different datasets.

~~Finally, the uncertainty range of the temperature trend of each dataset in different latitudes were compared. The uncertainty of every dataset is very small in the middle and low latitudes, and the largest uncertainty was in the high latitudes. In the high latitudes of the Southern Hemisphere, the uncertainty in CMST is the smallest. In the Northern Hemisphere high latitudes, the uncertainty in CMST is larger than BEST2 but smaller than other datasets.~~

**4.3 Inter-annual variations**

 8 shows the area weighted averaged time series of the global ST anomalies for the period . The temperature anomalies showed a clear warming trend from 1900 to

2017. For CMST, the highest temperature anomaly is 0.82 ℃ in 2016. There is a significant warming trend from 1910sfrom 1910s to 1940s and 1960s to 2017. In contrast, there is a cooling trend during 1940s to 1950s. The changes were highly consistent with the other datasets and are related to the changes of El Niño and La Niña events, volcanic eruptions, sea ice cover, and other factors (Simmons et al., 2017). Overall, the global ST changes in CMST and other datasets are similar over the period of 1900-2017. From 1920s to 1970s, CMST is slightly lower whereas HadCRUT4 is slightly higher than other datasets. The maximum difference in between CMST and HadCRUT4 is in 1938 and 1948, and the difference in temperature anomalies within these two years is 0.18 ℃. In 1938, the temperature anomalies are 0.17 ℃ and 0.01 ℃ in CMST and HadCRUT4, respectively. In 1948, the temperature anomalies are -0.20 ℃ and -0.02 ℃ in CMST and CRUT4, respectively.

The time series of ST anomalies in the seven datasets are also divided into the Northern Hemisphere (a), the Southern Hemisphere (b), and five latitudinal zones in 90 °N – 6 0 °N (c), 60 °N – 30 °N (e), 30 °N – 30 °S (g), 30 °S – 60 °S (f), and 60 °S – 90 °S (h). =Results clearly showed the time series of temperature anomalies in every dataset is highly consistent in the Northern Hemisphere (Figure 10a9a). At the low latitudes (Figure. 10e9e, f, g), the maximum ST of several datasets occurs in 2016, whereas the minimum occurs in different years. The minimum ST was found in 1917 for most of the datasets (GISTEMP1, GISTEMP2, BEST1, BEST2, HadCRUT4, and CMST), but it was found in 1908, 1909, and 1910 in NOAAGlobalTemp (Figure 10a9a).

In the mid latitude zone (Figure 10e9e, f), the times with maximum ST in seven datasets are generally consistent. The maximum ST occurs in 2015 in the 60 °N – 30 °N and in 2017 in the 30 °S – 60 ° S. The times with the minimum ST appear to be the same in seven datasets. The minimum ST wasST was detected in 1912 in the 60 °N – 30 °N, and in 1911 in the 30 °S – 60 °S. In the high latitudes of the Northern Hemisphere, the maximum ST consistently occurs in 2016, and the minimum ST consistently happened in 1902.

In the high latitudes of the Southern Hemisphere (Figure 10d9d), the CMST is consistent with all the series derived from other datasets after 1960. There are many less stations/grid boxes in the Antarctic/higher latitudes, hence larger variances were found before 1960.

**10.5. Data Availability**

The datasets used in CMST were derived from published data by NHMS (China, Russia, USA, Canada, Australia, some Asian countries, etc.) or climate data research institutions (UK/CRU, NOAA/NCEI). Part of the data are exchanged from some countries or regions, and therefore is conditionally available to public. Details of the data sources are as follows: The C-LSAT 1.3 in gridded form with a resolution of $5° × 5°$ developed by SUN Yat-Sen University (SYSU) & China Meteorological Administration is available on the Climate Explorer website of the Royal Netherlands Meteorological Institute (KNMI)

(http://climexp.knmi.nl/select.cgi?id=someone@somewhere&field=clsat_tavg). ERSST.v5 is from the

US                                    NOAA/NCEI                                    at

https://www.ncdc.noaa.gov/data-access/marineocean-data/extended-reconstructed-sea-surface-temperat

ure-ersst-v5. The China Merged Surface Temperature Data (CMST) dataset developed by SYSU is

currently public released on the Climate Explorer website of the Royal Netherlands Meteorological

Institute (KNMI) (http://climexp.knmi.nl/select.cgi?id=someone@somewhere&field=cmst ). With the

digital object identifiers (DOIs) (https://doi.pangaea.de/10.1594/PANGAEA.901295) issued for the data

sets (Yun et al., 2019), we hope to have provided a repository of a new global ST analyses across the

past 120 years from now to year 1900, for the scientific user community as well as the public.

**6.  Conclusion**

A new global ST dataset of CMST (China Merged Surface Temperature) has been developed based

on the LSAT dataset (C-LSAT1.3) and SST dataset (ERSSTv5). This dataset was completed by the

cooperation between  SUN Yat-sen University (SYSU), China Meteorological Administration

(CMA), and the United States NOAA/NCEI. In CMST, we found:

1) The spatial coverage  is larger when C-LSAT1.3 and ERSSTv5 are merged than that

from  merging C-LSAT1.3  with HadSST3 , particularly in the Polar

Regions. Besides that, the former (merging C-LSAT1.3 with ERSSTv5, named as CMST) is also

superior in terms of spatial distribution and temporal change with the reference series (derived from average of merged C-LSAT1.3 and four SST datasets).

2) The LSAT in CMST used a high-quality C-LSAT1.3. More than 6 900 stations were added to the previous version of C-LSAT1.0 (Xu et al., 2018), which has further increased the data coverage. The newly added stations are mainly from the ISTI dataset. The SST in CMST uses ERSSTv5 that using the ocean data from the latest ICOADS R3.0 and incorporates multiple types of observations. Compared with other existing global ST datasets, the CMST increases the overall coverage over global land and ocean surface.

3) The time series in CMST in the global and mid-low latitudes are  consistent with the other merged datasets for both inter-annual and inter-decadal timescales. In the high-latitude zones of NH and SH where the differences of temperature trends are usually larger, the trend of CMST grasped the major long-term climate changes features. Therefore, the CMST temperature trend from 1900 to 2017 is overall consistent with other datasets and proved to be a new useful tools in global climate change studies.

**Acknowledgement**: This research is supported by National Key R&D Program of China (Grant: 2017YFC1502301),

the Natural Science Foundation of China (Grant: 91546117), the Ministry of Science and Technology of China (Grant: GYHY201406016), and the China Postdoctoral Science Foundation (Grant: 2018M640848). We thank many contributors who contribute to the establishment of this dataset.

Table1. Comparison of the station number of the LSAT dataset during 1900 - 2017 (data length > 15 years)

| | C-LSAT | CRUTEM4 | GHCN |
|---|---|---|---|
| Global | **13687** | 9415 | 6871 |
| Northern Hemisphere (NH) | 11270 | 7881 | 5633 |
| Southern Hemisphere (SH) | **2418** | 1535 | 1238 |
| Africa | **922** | 749 | 586 |
| Asia | **2747** | 1831 | 1129 |
| Australia | **1022** | 388 | 563 |
| Europe | **3041** | 2177 | 930 |
| North America | 3462 | 2058 | 2699 |
| South America | **753** | 669 | 340 |
| Arctic | **1105** | 1050 | 278 |
| Antarctic | **104** | 36 | 36 |

Table 2. Current international marine dataset for climate change research

| Datasets | Resolution | Time | Mainly used observation data | Satellite data |
|----------|-----------|------|------------------------------|----------------|
| HadSST3 | $5°\times5°$ | 1850- | ICOADS R2.5 and some GTS data | No |
| ERSST.v5 | $2°\times2°$ | 1854- | ICOADS R3.0, Argo temperature above 5 m depth (Argo5obs), HadISST2 sea ice concentration, HadNMAT2, WOISST, Unadjusted SST | No |
| HadISST1 | $1°\times1°$ | 1870- | Met Office Marine Data Bank(MDB), GTS data (Since 1982), ICOADS (Use ICOADS SST data as a supplement in places where MDB data is missing) | Yes |
| COBE-SST2 | $1°\times1°$ | 1850- | ICOADS R2.5, MDB | Yes |

Table 32. Mean, max and min of monthly global coverage between 1900-01 and 2017-12

| Dataset | Coverage Mean | Coverage Max | Coverage Min |
|---------|---------------|--------------|--------------|
| Merge1  | 0.761         | 0.822        | 0.658        |
| Merge2  | 0.588         | 0.784        | 0.305        |

Table 43. Regional ST trends for different latitude zones from 1900 to 2017 ( °C/decade)

| | 90 °N – 60 °N | 60 °N – 30 °N | 30 °N – 30 °S | 30 °S – 60 °S | 60 °S – 90 °S |
|---|---|---|---|---|---|
| CMST | 0.116±0.012 | 0.098±0.006 | 0.082±0.005 | 0.080±0.003 | 0.046±0.004 |
| BEST1 | 0.149±0.016 | 0.104±0.006 | 0.071±0.005 | 0.090±0.003 | 0.113±0.008 |
| BEST2 | 0.118±0.010 | 0.102±0.006 | 0.071±0.005 | 0.086±0.003 | 0.055±0.005 |
| HadCRUT4 | 0.143±0.015 | 0.096±0.006 | 0.066±0.004 | 0.087±0.003 | 0.114±0.019 |
| GISTEMP1 | 0.142±0.013 | 0.090±0.006 | 0.077±0.004 | 0.085±0.002 | 0.037±0.006 |
| GISTEMP2 | 0.164±0.014 | 0.093±0.007 | 0.080±0.004 | 0.085±0.002 | 0.073±0.009 |
| NOAAGlobalTemp | 0.127±0.012 | 0.094±0.006 | 0.084±0.004 | 0.079±0.003 | 0.031±0.011 |

[Figure]

Figure 1. Stations added in C-LSAT version 1.3 between 1900 and 2017. The number on the right side of the color bar indicated the length of time and the number on the left side is the stations' number corresponding to a length of time.

(a)

[Figure]

(b)

[Figure]

(c)

[Figure]

Figure 2. The distribution of stations number from 1900 to 2017 for C-LSAT (a) and BE (b), and the comparisons of effective grid box numbers in North America (c).

[Figure]

Figure 32. Calculation method for temperature anomalies with a resolution of $5°\times5°$ for the grid with ocean and land

[Figure]

Figure 43. Comparison of monthly global coverage of the two datasets during 1900 to 2017. The grey line is Merge2 but using the original data.

[Figure]

Figure 54. Spatial distribution of 20-year average temperature anomalies between 1900 and 2017 in Merge1 (left) and Merge2 (right).

[Figure]

[Figure]

Figure 65. Differences between the merged series and the reference series during 1900 – 2017 in (left) 90 °N – 60 °N and (right) 60 °S – 90 °S. Blue line showed the difference between Merge1 and the reference series, and red line indicated the difference between Merge2 and the reference series.

(a)

[Figure]

(b)

Figure 76. Comparison of global ST dataset coverage between 1900 and 2017 (a) monthly coverage for all grid boxes; (b) annual average of coverage of monthly grid data.

[Figure]

Figure 87. Comparison of the annual averages of ST datasets coverage for each latitude zone between 1900 and 2017 in (a) 90 °N – 60 °N, (b) 60 °S – 90 °S, (c) 60 °N – 30 °N, (d) 30 °S – 60 °S and (e) 30 °N – 30 °S).

[Figure]

Figure 98. Comparison of global mean ST anomalies series during 1900-2017 for different datasets (relative to 1961-1990)

[Figure]

(g)

[Figure]

Figure 9. Comparison of regional ST anomalies series during 1900 – 2017 in (a) NH, (b) SH, (c) 90 °N – 60 °N, (d) 60 °S – 90 °S, (e) 60 °N – 30 °N, (f) 30 °S – 60 °S and (g) 30 °N – 30 °S.